# ELoRA: Low-Rank Adaptation for Equivariant GNNs

**Chen Wang** [1 2]  **Siyu Hu** [1 2]  **Guangming Tan** [1 2]  **Weile Jia** [1 2]

## Abstract

Pre-trained interatomic potentials have become a new paradigm for atomistic materials simulations, enabling accurate and efficient predictions across diverse chemical systems. Despite their promise, fine-tuning is often required for complex tasks to achieve high accuracy. Traditional parameter-efficient fine-tuning approaches are effective in NLP and CV. However, when applied to SO(3) equivariant pre-trained interatomic potentials, these methods will inevitably break equivariance—a critical property for preserving physical symmetries. In this paper, we introduce ELoRA (Equivariant Low-Rank Adaptation), a novel fine-tuning method designed specifically for SO(3) equivariant Graph Neural Networks (GNNs), the backbones in multiple pre-trained interatomic potentials. ELoRA adopts a path-dependent decomposition for weights updating which offers two key advantages: (1) it preserves SO(3) equivariance throughout the fine-tuning process, ensuring physically consistent predictions, and (2) it leverages low-rank adaptations to significantly improve data efficiency. We prove that ELoRA maintains equivariance and demonstrate its effectiveness through comprehensive experiments. On the rMD17 organic dataset, ELoRA achieves a 25.5% improvement in energy prediction accuracy and a 23.7% improvement in force prediction accuracy compared to full-parameter fine-tuning. Similarly, across 10 inorganic datasets, ELoRA achieves average improvements of 12.3% and 14.4% in energy and force predictions, respectively. Code will be made publicly available at https://github.com/hyjwpk/ELoRA.

## 1. Introduction

Deep learning-based potentials hold significant promise in addressing the accuracy-efficiency trade-off in DFT calculations and can expand the spatial and temporal scales of molecular dynamics simulations (Gong et al., 2023). A variety of deep learning potentials (Behler & Parrinello, 2007; Smith et al., 2017; Schütt et al., 2017; Gilmer et al., 2017; Zhang et al., 2018; Unke & Meuwly, 2019; Unke et al., 2021; Klicpera et al., 2021) have demonstrated the ability to predict total energies and atomic forces for given material structures. Recently, with the continuous generation of ab initio datasets and a deeper understanding of symmetry, pre-trained potentials have emerged in large numbers, such as MACE (Batatia et al., 2022b), M3GNet (Chen & Ong, 2022), Equiformerv2 (Liao et al., 2023), Uni-Mol (Zhou et al., 2023), CHGNet (Deng et al., 2023), SevenNet (Park et al., 2024), GPTFF (Xie et al., 2024), and Orb (Neumann et al., 2024). Among these, equivariant Graph Neural Networks (GNNs) stand out as particularly effective for modeling physical systems. By leveraging equivariance, these models achieve high performance, even with limited first-principles data (Satorras et al., 2021; Batzner et al., 2022; Batatia et al., 2022b; Pelaez et al., 2024). This is particularly valuable because generating first-principles training data is often extremely resource intensive. Equivariant pre-trained models have been increasingly employed in research in finding symmetry-breaking order parameters (Smidt et al., 2021) and predicting molecular properties (Miller et al., 2020), phonon density of states (Chen et al., 2021), etc.

However, due to the diversity of material structures and the significant variations of atomic interactions across different materials, pre-trained interatomic potentials may not achieve the desired accuracy when dealing with material structures that are not well learned during training. Moreover, obtaining accurate labels for such configurations typically requires costly first-principles calculations, which further limits the diversity of data available for training. These potentials can only be applied to general downstream tasks without fine-tuning. For example, the accuracy of pre-trained MACE-MP (Batatia et al., 2023) is insufficient to study molecular crystal polymorphs (Kaur et al., 2025).

Fine-tuning the pre-trained model with downstream data helps refine the interatomic potential to better predictions.

---

[1]State Key Lab of Processors, Institute of Computing Technology, Chinese Academy of Sciences [2]University of Chinese Academy of Sciences. Correspondence to: Siyu Hu <husiyu@ict.ac.cn>, Guangming Tan <tgm@ict.ac.cn>, Weile Jia <jiaweile@ict.ac.cn>.

*Proceedings of the $42^{nd}$ International Conference on Machine Learning*, Vancouver, Canada. PMLR 267, 2025. Copyright 2025 by the author(s).

Fine-tuning approaches can be broadly categorized into full-parameter fine-tuning and parameter-efficient fine-tuning (PEFT) (Lialin et al., 2023). Full-parameter fine-tuning has been explored in pre-trained interatomic potentials like MACE (Kaur et al., 2025). However, as pre-trained models continue to grow in size, full-parameter fine-tuning becomes increasingly computationally expensive and memory intensive. Moreover, it may cause catastrophic forgetting of pre-trained knowledge during fine-tuning (Kirkpatrick et al., 2017). In contrast, PEFT methods are designed to modify only a small subset of the model's parameters and keep most parameters fixed. This approach reduces computational overhead and memory usage compared to full-parameter fine-tuning. Furthermore, PEFT methods help mitigate catastrophic forgetting, improving model transferability and generalization (Ding et al., 2023). PEFT methods, such as adapter-based fine-tuning (Houlsby et al., 2019) and low-rank adaptation (LoRA) (Hu et al., 2022), have been developed and demonstrated superior advantages in terms of resource efficiency and scalability. However, these methods are developed primarily for transformer-based models. For GNNs, specialized methods such as AdapterGNN (Li et al., 2024) have been introduced to adapt pre-trained GNN models to downstream tasks. Unfortunately, these methods cannot be directly applied to SO(3) equivariant GNNs, as they will destroy the equivariance during fine-tuning. To the best of our knowledge, no PEFT methods have been specifically designed for SO(3) equivariant GNNs.

This highlights the need for SO(3) equivariant GNN-based PEFT methods, as SO(3) equivariant GNN pre-trained models present a promising path for modeling universal interatomic potentials and play a crucial role in material and chemistry simulations.

Our contributions can be summarized as follows:

- We explore the difference between two dominant interatomic potential training paradigms (From-Scratch-Training Paradigm and Pre-training-Fine-tuning Paradigm) from the perspective of chemical space. We point out that Pre-training-Fine-tuning Paradigm exhibits stronger generalization capabilities.

- We develop a PEFT strategy tailored for SO(3) equivariant GNN models, called ELoRA (Equivariant Low-Rank Adaptation). We prove that ELoRA can preserve the equivariance during fine-tuning.

- The experiments show that ELoRA works both on organic and inorganic pre-trained models. For example, on the rMD17 organic dataset, ELoRA achieves 25.5% and 23.7% improvement in energy and force predictions compared to full-parameter fine-tuning. On 10 inorganic real cases, the average improvements are 12.3% and 14.4%.

## 2. Preliminary

**LoRA (Low-Rank Adaptation)**: Low-Rank Adaptation (LoRA) (Hu et al., 2022) is a PEFT technique for Large Language Model (LLM). The core idea behind LoRA is that the updates to the pre-trained model weights, denoted as $W_0 \in \mathbb{R}^{d \times k}$, during fine-tuning exhibit a low intrinsic rank. As a result, the update $\Delta W$ can be approximated as the product of two low-rank matrices $B \in \mathbb{R}^{d \times r}$ and $A \in \mathbb{R}^{r \times k}$, where $r$ is the rank and $r \ll \min(d, k)$. The forward pass $h = W_0 x$ then be modified as:

$$h = W_0 x + \Delta W x = W_0 x + \frac{\alpha}{r} BA x, \qquad (1)$$

where $\alpha$ is a scaling constant, $A$ is initialized with a random Gaussian distribution and $B$ is zero-initialized, so $\Delta W$ is zero at the beginning of training. LoRA reduces the weights that need to be updated during training. During inference, $\Delta W$ can be merged into $W_0$, ensuring there is no additional latency compared to a full-parameter fine-tuned model.

**SO(3) EMPNNs (Equivariant Message Passing Neural Networks)**: Equivariance is introduced by (Cohen & Welling, 2016) to maintain physical symmetries of the input data. They use geometric tensors as node embeddings and ensure equivariance by imposing constraints on the operations that can be performed (Kondor et al., 2018). Formally, a mapping $f : X \to Y$ is equivariant for vector spaces $X$ and $Y$ to a group $G$, if $f \circ D_X(g) = D_Y(g) \circ f$ is satisfied for any $g$ in $G$, where $D_X$ and $D_Y$ represent the transformation of group $G$ on vector spaces $X$ and $Y$, respectively. More detailed information on group theory and equivariance is displayed in Appendix A. SO(3) equivariant GNNs leverage the inherent symmetry of their domains with respect to SO(3), the group of 3D rotations. They transmit equivariant messages between nodes (representing atoms) through edges (representing bonds or interactions between atoms) in the graph. This approach is commonly referred to as SO(3) Equivariant Message Passing Neural Networks (EMPNNs). Tensor field networks (Thomas et al., 2018), NequIP (Batzner et al., 2022), MACE (Batatia et al., 2022b), Allegro (Musaelian et al., 2023) and Equiformerv2 (Liao et al., 2023) belong to EMPNNs. Among them, MACE (Batatia et al., 2022b) is a representative SO(3) EMPNN model that uses higher body order messages.

**SH (Spherical Harmonics) and TP (Tensor Product)**: In SO(3) EMPNNs, node features are represented using the coefficients of spherical harmonics, ensuring SO(3) equivariance. Spherical harmonics, denoted as $Y_m^l$, are functions that map points on the sphere to real or complex numbers, where $l$ is a non-negative integer and $m$ is an integer ranging from $-l$ to $l$. The features of a node $i$ are represented as $h_{i,klm}$, where $l$ is the rotation order with $0 \leq l \leq L$, $m$ ranges from $-l$ to $l$, and $k$ indicates the channel index with

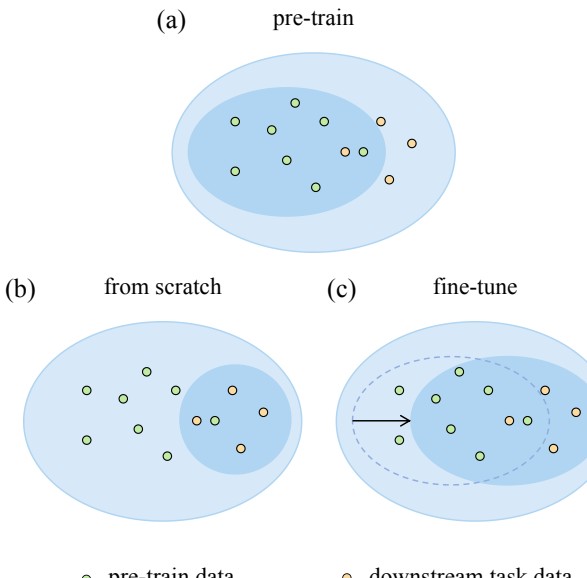

(a) pre-train

(b) from scratch  (c) fine-tune

○ pre-train data    ○ downstream task data

*Figure 1.* **Comparison of chemical space covered by pre-trained (a), from scratch (b), and fine-tuned (c) model.** The dark blue part of the figure represents the chemical space covered by the model. The fine-tuned model covers more chemical space than the model trained from scratch, as it retains knowledge inherited from the pre-trained model.

$1 \leq k \leq K_l$. The generalized tensor product is an equivariant operation defined using Clebsch-Gordan coefficients $C$ (Griffiths & Schroeter, 2019) as:

$$(u \otimes v)_{l_3 m_3} = \sum_{l_1 m_1, l_2 m_2} C^{l_3 m_3}_{l_1 m_1, l_2 m_2} u_{l_1 m_1} v_{l_2 m_2}. \quad (2)$$

This tensor product produces non-zero values only for $|l_1 - l_2| \leq l_3 \leq (l_1 + l_2)$ ($m_3$ ranges from $-l_3$ to $l_3$). Due to this restriction, the tensor product can be divided into multiple different paths by $(l_1, l_2, l_3)$, only the path $(l_1, l_2, l_3)$ that satisfies $|l_1 - l_2| \leq l_3 \leq (l_1 + l_2)$ is non-zero.

## 3. Fine-tuning Covers a Larger Chemical Space

Pre-trained models tend to achieve better generalization than task-specific models because they are trained on larger and more diverse datasets. For pre-trained interatomic potentials, their generalization ability is reflected in covering a wider range of chemical space, as Figure 1(a) shows. In contrast, task-specific models trained from scratch tend to have limited generalization capabilities and cover a smaller range of chemical space, as shown in Figure 1(b). These models learn exclusively from the dedicated training set of the downstream task.

Even though the pre-trained models demonstrate superior generalization capabilities, they may not be sufficiently accurate to make predictions on certain out of domain (OOD) data in complex downstream tasks. Pre-training datasets at first-principles accuracy are often sparse across the full chemical space. This is primarily due to the high computational cost of generating such data using density functional theory (DFT), as well as the combinatorial complexity of chemical systems. This inherent sparsity presents a challenge to the accuracy of pre-trained interatomic potentials.

Fine-tuning is a commonly used technique to improve the performance of pre-trained models on specific tasks. Fine-tuning transfers knowledge from a large pre-trained model to a more specific task. Through fine-tuning, the performance of the model in specific tasks or compounds is enhanced, while its ability to generalize across a broader chemical space is also retained (depicted in Figure 1(c)). As observed in Section 5, when trained on the same task-specific dataset, the fine-tuned pre-trained model often achieves better performance on the test set compared to a model trained from scratch, indicating improved generalization and data efficiency. Moreover, generating training data with *ab initio* accuracy is computationally costly due to the need for density functional theory (DFT) calculations, which makes the data efficiency of fine-tuning particularly valuable in practical applications.

## 4. Equivariant PEFT

EMPNNs can be broken into three main modules: input embedding, interaction blocks, and readout layer. Interaction blocks update the node features iteratively which are the main part of EMPNNs. Notably, the majority of the trainable parameters in EMPNNs are concentrated in the interaction blocks. For example, in the pre-trained model, MACE-MP (Batatia et al., 2023), approximately 99.7% of the model weights reside in the interaction blocks. Each interaction block is composed of three fundamental operations: point convolution (Thomas et al., 2018), self-interaction (Thomas et al., 2018), and residual connection (He et al., 2016). We prove that these key operations can be induced into a unified tensor-product (TP) formalism, which enables a consistent PEFT method for implementing equivariance. The TP transformation is introduced in 4.1. The implementation of ELoRA is described in 4.2.

### 4.1. TP transformation

We first make a statement about the three key operations and prove that they can be induced into a unified TP formalism.

**Point Convolution**: Message passing between adjacent nodes is performed using point convolution. This operation is applied to each node based on its neighboring nodes. To

ensure that the point convolution remains equivariant, the edges are embedded in the following form to derive the rotation equivariant filter: $F_{kl_1m_1}^{l_2l_3}(\vec{r}) = R_{kl_1l_2l_3}(r)Y_{m_1}^{l_1}(\vec{r})$, where $R_{k_1l_1l_2l_3}(r)$ is a learnable function and $Y_{m_1}^{l_1}(\vec{r})$ is the spherical harmonic. When computing the point convolution centered at node $i$, the contribution of node $j$ is:

$$m_{ij,kl_3m_3} = \sum_{l_1m_1,l_2m_2} F_{kl_1m_1}^{l_2l_3}(\vec{r_{ji}}) \otimes h_{j,kl_2m_2}, \quad (3)$$

where $F_{kl_1m_1}^{l_2l_3}(\vec{r_{ji}})$ and $h_{j,kl_2m_2}$ are considered as tensors and $\otimes$ denotes the tensor product. $m_{ij,kl_3m_3}$ is the message passing between nodes $i$ and $j$. By summing $m_{ij,kl_3m_3}$ over all the neighboring nodes $\mathcal{N}(i)$, we can obtain the messages of node $i$:

$$m_{i,kl_3m_3} = \sum_{j\in\mathcal{N}(i)} m_{ij,kl_3m_3}. \quad (4)$$

**Self-interaction**: The self-interaction layer mixes the channels of features at each point which is analogous to $1\times 1$ convolutions:

$$\sum_{\tilde{k}} W_{k\tilde{k}l}h_{\tilde{k}lm}, \quad (5)$$

where $W_{k\tilde{k}l}$ is the learnable weight.

**Residual connection**: The residual connection layer performs point convolution on the initial feature of the node (chemical species $z_i$) and the current node feature $h_i$:

$$\sum_{\tilde{k}} W_{z_ikl,\tilde{k}}h_{i,\tilde{k}lm}, \quad (6)$$

where $W_{z_ikl,\tilde{k}}$ is the learnable weight, and the result is added to the residual of the current layer as the feature update.

**Fully Connected Tensor Product**: For tensors $u$ and $v$ with multiple channels, the fully connected tensor product is defined to mix different channels. Specifically, $(u \otimes v)_{k_3l_3m_3}$ is defined as:

$$\sum_{k_1l_1m_1,k_2l_2m_2} W_{k_3k_2k_1,l_3l_2l_1}C_{l_1m_1,l_2m_2}^{l_3m_3}u_{k_1l_1m_1}v_{k_2l_2m_2}, \quad (7)$$

where $W_{k_3k_2k_1,l_3l_2l_1}$ is the learnable weight corresponding to the path $(l_1, l_2, l_3)$.

**Proposition 4.1.** *The self-interaction of the tensor $h$ is equivalent to the fully connected tensor product of $h$ and $1$.*

**Proposition 4.2.** *The residual connection of the tensor $h_i$ is equivalent to the fully connected tensor product of $h_i$ and $\hat{z}_i$, where $\hat{z}_i$ is the one-hot encoding of the chemical species $z_i$.*

The proofs of the propositions are given in Appendix C. Proposition 4.1 and Proposition 4.2 point out that the formalization of the self-interaction and residual connection are

special cases of the fully connected tensor product. Therefore, the three key operations can be induced into a unified TP formalism. In Section 4.2, we focus on introducing ELoRA for the fully connected tensor product.

## 4.2. ELoRA: Equivariant LoRA

We propose a novel path-dependent low-rank adaptation in SO(3) EMPNNs' fine-tuning process. Then we prove that ELoRA can project equivariant messages into low-dimensional space while preserving the inherent equivariance.

**A path-dependent decomposition**: The equivariance of the tensor product stems from its constraints on paths. To preserve equivariance, we employ a path-dependent decomposition for weight updates, as opposed to the global decomposition used in LoRA. This approach assumes that the weights along each path exhibit a lower intrinsic dimension during adaptation. Formally, for a pre-trained weight matrix $W^0$ utilized in a fully connected tensor product, we constrain its updates by decomposing the updates along each path into a low-rank representation:

$$W_{l_3l_2l_1}^0 + \Delta W_{l_3l_2l_1}^0 = W_{l_3l_2l_1}^0 + B_{l_3l_2l_1}A_{l_3l_2l_1}, \quad (8)$$

where $W_{l_3l_2l_1}^0 \in \mathbb{R}^{K_{l_3}^3 \times (K_{l_2}^2 \cdot K_{l_1}^1)}$, $B_{l_3l_2l_1} \in \mathbb{R}^{K_{l_3}^3 \times R}$, $A_{l_3l_2l_1} \in \mathbb{R}^{R \times (K_{l_2}^2 \cdot K_{l_1}^1)}$ and the rank $R \ll \min(K_{l_3}^3, K_{l_2}^2 \cdot K_{l_1}^1)$. $K$ is the number of channels, and the superscript of $K$ indicates whether it is associated with the first (1) or second (2) input tensor or the output tensor (3).

We illustrate our approach in Figure 2. Taking tensors $u$ and $v$ with rotation order $0 \le l \le 1$ as an example, Figure 2(a) demonstrates the fully connected tensor product of $u$ and $v$ using pre-trained weights $W^0$. In this process, $u$ and $v$ first perform a generalized tensor product, but only compute the results for each path without calculating the output tensor, after which $W^0$ combines the results from different paths into the output tensor. Figure 2(b) shows the corresponding computation process using ELoRA weights $A$ and $B$. In contrast to the fully connected tensor product, this approach replaces the fully connected operation with a two-step process: first, project along the paths using $A$, and then fully connect the results to the output tensor using $B$. The result of the fully connected tensor product of $u$ and $v$ using ELoRA fine-tuned weights is the sum of the results from Figure 2(a) and Figure 2(b).

We give the following propositions to explain how ELoRA projects equivariant messages into low-dimensional space while preserving the inherent equivariance. The proofs are detailed in Appendix C.

**Proposition 4.3.** *For the pre-trained weight matrix $W^0$ of the fully connected tensor product and its corresponding ELoRA matrices $A$ and $B$, the fully connected tensor prod-*

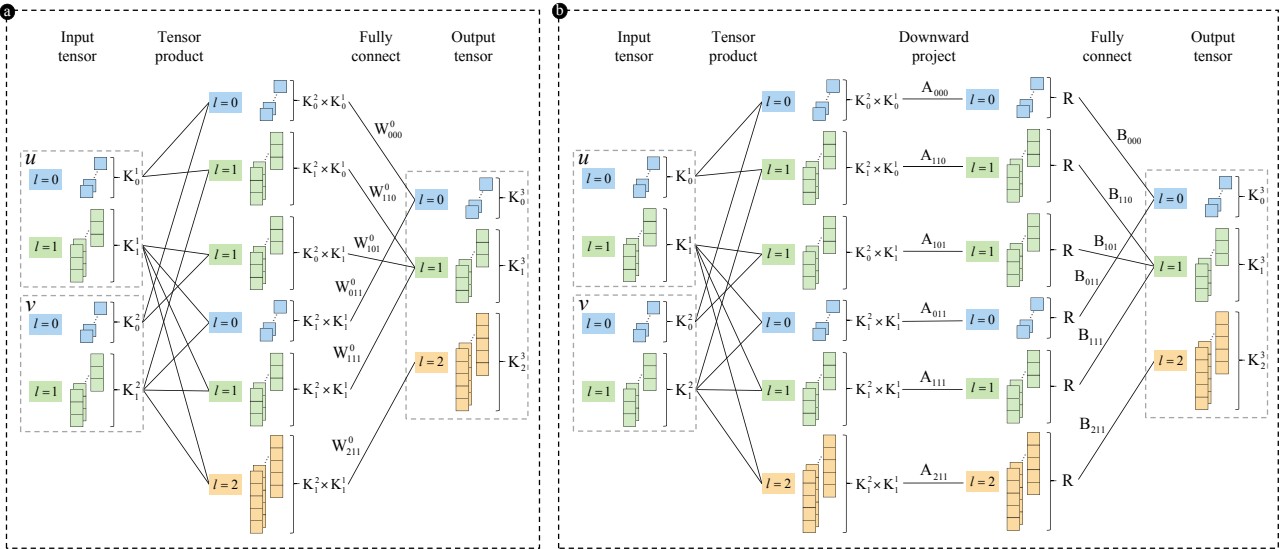

Figure 2. **An overview of ELoRA.** (a) the fully connected tensor product of tensors $u$ and $v$ using pre-trained weights $W^0$. (b) the corresponding computation process using ELoRA weights $A$ and $B$. The final result is the sum of the results of (a) and (b).

*uct of the tensors $u$ and $v$ with ELoRA is SO(3) equivariant.*

Proposition 4.3 demonstrates that the overall rotational equivariance is preserved even after incorporating ELoRA. We then present the following proposition to illustrate how the path-dependent decomposition employed by ELoRA projects the equivariant messages, as depicted in Figure 2.

**Proposition 4.4.** *For the pre-trained weight matrix $W^0$ of the fully connected tensor product and its corresponding ELoRA matrices $A$ and $B$, the fully connected tensor product of the tensors $u$ and $v$ with ELoRA satisfies $(u \otimes v)_{k_3 l_3 m_3} = (u \otimes v)^0_{k_3 l_3 m_3} + \sum_{l_1, l_2} \sum_r B_{k_3 r, l_3 l_2 l_1} (u_{l_1} \otimes v_{l_2})_{r l_3 m_3}$, where $(u \otimes v)^0_{k_3 l_3 m_3}$ is the fully connected tensor product of tensors $u$ and $v$ using $W^0$ without ELoRA and $(u_{l_1} \otimes v_{l_2})_{r l_3 m_3}$ is the fully connected tensor product of tensors $u_{l_1}$ and $v_{l_2}$ using ELoRA matrix $A$.*

Proposition 4.4 states that, after applying ELoRA, the fully connected tensor product of the tensors $u$ and $v$ can be decomposed into two components. The first component is the fully connected tensor product computed using $W^0$. The second component involves using $A_{l_3 l_2 l_1}$ to project the message along the path $(l_1, l_2, l_3)$ onto $(u_{l_1} \otimes v_{l_2})_{r l_3 m_3}$, where the projected $(u_{l_1} \otimes v_{l_2})_{r l_3 m_3}$ has its channels reduced to $R$. This projection is then followed by a fully connected operation applied to the projected message using $B_{l_3 l_2 l_1}$, which produces the output tensor. Therefore, we have the following corollary.

**Corollary 4.5.** *For the pre-trained weight matrix $W^0$ of the fully connected tensor product and its corresponding ELoRA*

*matrices $A$ and $B$, $A_{l_3 l_2 l_1}$ projects the equivariant message on the path $(l_1, l_2, l_3)$ in the fully connected tensor product of the tensors $u$ and $v$ with ELoRA into a low-dimensional space of dimension $R$.*

## 5. Experiments

### 5.1. Experimental Setup

#### 5.1.1. PRE-TRAINED MODELS

MACE (Batatia et al., 2022b) is a fundamental EMPNN that utilizes high body order messages and achieves state-of-the-art prediction performance. We adopt the MACE architecture as the backbone to evaluate the effectiveness of ELoRA. Details of the MACE architecture are provided in Appendix B. Due to the distinct properties of organic and inorganic systems, the pre-trained models are divided into two classes, the organic pre-trained model, MACE-OFF (Kovács et al., 2023) and the inorganic pre-trained model, MACE-MP (Batatia et al., 2023). MACE-OFF is pre-trained on SPICE (Eastman et al., 2023), while MACE-MP is pre-trained on MPTrj (Deng et al., 2023). More details about the pre-trained datasets can be found in Appendix E.1.

#### 5.1.2. DATASETS

For organic downstream tasks, we use the revised MD17 (rMD17) (Christensen & Von Lilienfeld, 2020), 3BPA (Kovács et al., 2021), and AcAc (Batatia et al., 2022a) datasets, which are representative benchmarks for organic systems. For inorganic downstream tasks, we employ 10

datasets that reflect a variety of real-world scenarios. These datasets are based on simulations of diverse inorganic compounds under complex temperature and pressure conditions, encompassing a wide range of chemical and configurational spaces. They include datasets for metal elements (Cu (Zhang et al., 2020), Sn (Chen et al., 2023), Ti (Wen et al., 2021), V (Wang et al., 2022a), W (Wang et al., 2022b), AgAu (Wang et al., 2021), AlMgCu (Jiang et al., 2021)), water/ice systems (Zhang et al., 2021), oxides (HfO2 (Wu et al., 2021)), and solid-state electrolytes (SSE-PBE (Huang et al., 2021)). These datasets are generated using the DP-GEN method (Zhang et al., 2020). Details of these datasets are given in Appendix E.2.

### 5.1.3. BASELINE

We compare the accuracy of models trained from scratch, pre-trained models, pre-trained models with full-parameter fine-tuning, and pre-trained models with ELoRA fine-tuning. The baseline for models trained from scratch includes ACE (Kovács et al., 2021), NequIP (Batzner et al., 2022), BOTNet (Batatia et al., 2022a), Allegro (Musaelian et al., 2023), DPA2 (Zhang et al., 2023), PACE (Xu et al., 2024), and MACE (Batatia et al., 2022b), all of which have demonstrated strong performance across various downstream tasks. The baseline for pre-trained models consists of the opensource organic MACE-OFF and inorganic MACE-MP. Both full-parameter fine-tuning and ELoRA fine-tuning are applied to these same pre-trained models (MACE-OFF and MACE-MP) under identical hyperparameter settings. More details of these models are provided in Appendix E.3.

### 5.2. Main Experimental Results

#### 5.2.1. RESULTS ON THE ORGANIC DATASETS

**Revised MD17**: To verify whether fine-tuning pre-trained models offers an advantage on downstream tasks with limited data, we follow the previous setting (Batatia et al., 2022b), using only 50 configurations for each organic molecule in the rMD17 dataset for training. As shown in Table 1, fine-tuning on MACE-OFF achieves higher accuracy than training from scratch. This is consistent with the analysis in Section 3, which indicates that fine-tuning will cover a broader chemical space in real cases. Furthermore, ELoRA fine-tuning outperforms full-parameter fine-tuning across all organic molecules listed in Table 1, achieving new state-of-the-art results. Compared to full-parameter fine-tuning, ELoRA improves energy prediction accuracy by 25.5% and force prediction accuracy by 23.7%.

**3BPA**: Compared to the rMD17 task with limited data, the 3BPA dataset is used to evaluate the model's extrapolation capabilities under varying temperatures and dihedral slices. The models are trained using datasets collected at 300 K, and the temperatures of the test set range from 300 K to 1200

*Table 1.* **MAE of energy (E, meV) and force (F, meV/Å) on the rMD17 dataset.** Each model is trained using only 50 molecules. The best results are in bold.

| Method | | ACE | NequIP | PACE | MACE | MACE | MACE |
|---|---|---|---|---|---|---|---|
| | | From scratch | | | | Full-parameter | ELoRA |
| Aspirin | E | 26.2 | 19.5 | 15.7 | 17.0 | 10.3 | **7.3** |
| | F | 63.8 | 52.0 | 37.4 | 43.9 | 23.6 | **17.6** |
| Azobenzene | E | 9.0 | 6.0 | 6.7 | 5.4 | 4.7 | **4.0** |
| | F | 28.8 | 20.0 | 17.5 | 17.7 | 15.1 | **12.4** |
| Benzene | E | **0.2** | 0.6 | 0.6 | 0.7 | 0.4 | **0.2** |
| | F | 2.7 | 2.9 | 3.3 | 2.7 | 2.4 | **1.6** |
| Ethanol | E | 8.6 | 8.7 | 6.3 | 6.7 | 2.7 | **2.1** |
| | F | 43.0 | 40.2 | 25.4 | 32.6 | 13.9 | **10.7** |
| Malonaldehyde | E | 12.8 | 12.7 | 11.5 | 10.0 | 7.3 | **6.5** |
| | F | 63.5 | 52.5 | 57.3 | 43.3 | 25.3 | **21.7** |
| Naphthalene | E | 3.8 | 2.1 | 2.1 | 2.1 | 1.8 | **1.4** |
| | F | 19.7 | 10.0 | 9.7 | 9.2 | 7.8 | **6.0** |
| Paracetamol | E | 13.6 | 14.3 | 10.1 | 9.7 | 6.9 | **4.8** |
| | F | 45.7 | 39.7 | 29.3 | 31.5 | 20.4 | **14.8** |
| Salicylic acid | E | 8.9 | 8.0 | 7.0 | 6.5 | 4.2 | **3.2** |
| | F | 41.7 | 35.0 | 29.2 | 28.4 | 17.8 | **14.2** |
| Toluene | E | 5.3 | 3.3 | 2.7 | 3.1 | 1.9 | **1.3** |
| | F | 27.1 | 15.1 | 12.0 | 12.1 | 8.6 | **5.9** |
| Uracil | E | 6.5 | 7.3 | 5.9 | 4.4 | 2.6 | **2.1** |
| | F | 36.2 | 40.1 | 26.8 | 25.9 | 14.7 | **11.6** |

K. The results (in Table 2) of MACE without fine-tuning are significantly less accurate than those obtained with fine-tuning, highlighting the necessity of fine-tuning pre-trained models. Furthermore, ELoRA fine-tuning outperforms training from scratch and achieves a new state-of-the-art. As the test set temperature increases from 300 K to 1200 K, the accuracy gap between the ELoRA fine-tuned model and the full-parameter fine-tuned model grows from 0.3 (meV) to 2.7 (meV) for energy and from 0.3 (meV/Å) to 6.7 (meV/Å) for forces. This demonstrates that the ELoRA fine-tuned model exhibits superior out-of-domain generalization capabilities.

In Figure 3, we compare the predictions of models trained on the 3BPA dataset for three dihedral slices. Overall, the results show that the ELoRA fine-tuned model aligns more closely with the ground truth. Notably, in the middle panel, which includes geometries farthest from the training set, the ELoRA fine-tuned model significantly outperforms the model trained from scratch, demonstrating remarkable extrapolation capabilities.

**AcAc**: The AcAc dataset is designed to evaluate a model's ability to extrapolate on the acetylacetone molecule under high temperatures, bond breaking, and bond torsion. As shown in Table 3, the pre-trained model MACE-OFF, when used without fine-tuning, performs significantly worse than the model trained from scratch. ELoRA fine-tuning outperforms training from scratch and full-parameter fine-tuning in all cases, achieving improvements of 26.4% and 14.4% in energy and force accuracy, respectively. After fine-tuning with ELoRA, the fine-tuned parameters can be merged into

*Table 2.* **RMSE of energy (E, meV) and force (F, meV/Å) on the 3BPA dataset.** The training set is collected at 300 K. Standard deviations are computed over three runs and shown in brackets. The best results are in bold.

|  |  | Allegro | NequIP | BOTNet | PACE | MACE | MACE | MACE | MACE |
|---|---|---|---|---|---|---|---|---|---|
| Method |  | From scratch | | | | | Full-parameter | ELoRA | MACE-OFF |
| 300 K | E | 3.84 (0.08) | 3.3 (0.1) | 3.1 (0.13) | **2.4 (0.1)** | 3.0 (0.2) | 3.3 (0.03) | 3.0 (0.05) | 17340.0 |
|  | F | 12.98 (0.17) | 10.8 (0.2) | 11.0 (0.14) | 9.1 (0.1) | 8.8 (0.3) | 7.8 (0.01) | **7.5 (0.05)** | 124.7 |
| 600 K | E | 12.07 (0.45) | 11.2 (0.1) | 11.5 (0.6) | 7.9 (0.2) | 9.7 (0.5) | 7.3 (0.04) | **6.5 (0.10)** | 17369.8 |
|  | F | 29.17 (0.22) | 26.4 (0.1) | 26.7 (0.29) | 21.4 (0.3) | 21.8 (0.6) | 16.6 (0.05) | **15.5 (0.12)** | 136.1 |
| 1200 K | E | 42.57 (1.46) | 38.5 (1.6) | 39.1 (1.1) | 29.6 (0.4) | 29.8 (1.0) | 20.3 (0.17) | **17.6 (0.11)** | 17412.8 |
|  | F | 82.96 (1.77) | 76.2 (1.1) | 81.1 (1.5) | 60.7 (2.0) | 62.0 (0.7) | 48.7 (0.56) | **42.0 (0.51)** | 157.3 |
| Dihedral Slices | E | - | - | 16.3 (1.5) | 7.6 (0.4) | 7.8 (0.6) | 7.3 (0.28) | **5.9 (0.28)** | 17309.4 |
|  | F | - | - | 20.0 (1.2) | 16.0 (0.5) | 16.5 (1.7) | 12.3 (0.10) | **11.4 (0.17)** | 116.2 |

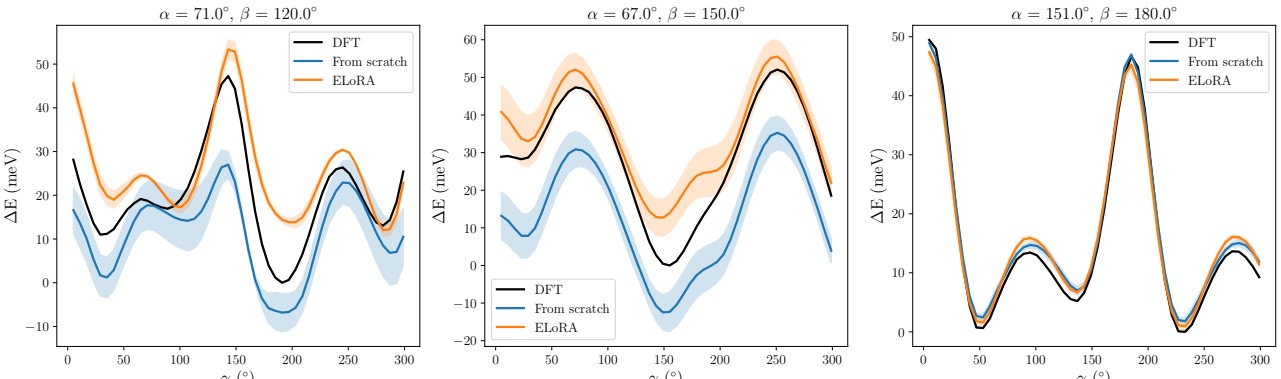

*Figure 3.* **Energy predictions for the three cuts of the potential energy surface of the 3BPA molecule by models trained from scratch and fine-tuned with ELoRA.** The true energy (DFT) is shown in black. For each cut, the curves have been vertically shifted so that the minimum true energy is zero. The shaded area represents the standard deviation.

the original model parameters during inference, adding no additional computational overhead. The reduction in the number of parameters in the fine-tuned model results from removing the weights associated with those elements of the pre-trained model that are absent from the downstream task.

### 5.2.2. RESULTS ON INORGANIC DATASETS

In this section, we evaluate the performance of ELoRA fine-tuning on inorganic datasets. Inorganic crystals are much more diverse than small organic molecules, and the 10 datasets used in this section are generated under complex temperature and pressure conditions, requiring models to demonstrate strong generalization capabilities. Model performance is evaluated using RMSE, with the results summarized in Table 4.

The results show that pre-trained models MACE-MP, when used without fine-tuning, significantly underperform models trained from scratch in inorganic datasets. However,

ELoRA fine-tuned pre-trained models outperform other models in force prediction on 9 out of 10 datasets. For energy prediction, the fine-tuned pre-trained models exceed the performance of the models trained from scratch using the same backbone, but are slightly behind state-of-the-art, likely due to differences in model architecture. Compared to full-parameter fine-tuning, ELoRA achieves an average improvement of 12.3% in energy prediction and 14.4% in force prediction across the 10 inorganic datasets.

### 5.3. Other Experimental Results

#### 5.3.1. DATA EFFICIENCY

Data efficiency can be measured by the size of the required training data. Taking rMD17-aspirin as an example, Figure 4 describes the comparison of full-parameter fine-tuning and ELoRA on the pre-trained model. ELoRA obtains a lower error than full-parameter fine-tuning when the training data size varies from 50 to 1000. ELoRA shows lower error than

*Table 3.* **RMSE of energy (E, meV) and force (F, meV/Å) on the acetylacetone dataset.** The training set is collected at 300 K. Standard deviations are computed over three runs and shown in brackets. The best results are in bold.

| | | BOTNet | NequIP | MACE | MACE | MACE | MACE |
|---|---|---|---|---|---|---|---|
| Method | | From scratch | | | Full-parameter | ELoRA | MACE-OFF |
| 300 K | E | 0.89 (0.0) | 0.81 (0.04) | 0.9 (0.03) | 1.0 (0.02) | **0.8 (0.03)** | 24183.8 |
| | F | 6.3 (0.0) | 5.90 (0.38) | 5.1 (0.10) | 5.1 (0.07) | **4.5 (0.06)** | 463.8 |
| 600 K | E | 6.2 (1.1) | 6.04 (1.26) | 4.6 (0.3) | 5.8 (0.28) | **3.9 (0.33)** | 24160.1 |
| | F | 29.8 (1.0) | 27.8 (3.29) | 22.4 (0.9) | 16.4 (0.70) | **13.6 (0.26)** | 474.2 |
| N° Parameters | | 2,756,416 | 3,190,488 | 2,803,984 | 751,896 | 751,896 | 1,428,368 |

full-parameter fine-tuning when the size of the downstream data is smaller. In cases of small data size, the amount of data required for fine-tuning can be reduced by 42% and 44% in reaching similar errors, thereby enhancing data efficiency. As the training data increases, the difference in error between full-parameter fine-tuning and ELoRA gradually decreases. Theoretically, their accuracy gap will diminish as the dataset size approaches infinity. Detailed theoretical analysis can be found in Appendix D.

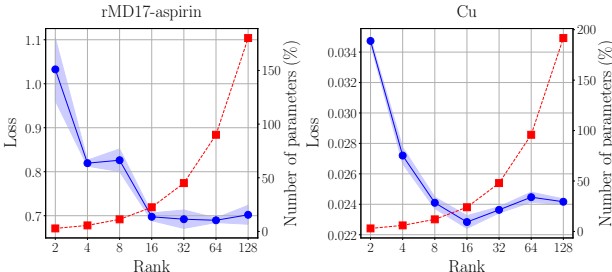

*Figure 5.* **The validation set loss and ELoRA parameter proportion across different ranks.** The results are obtained from tests conducted on the rMD17-aspirin and inorganic-Cu datasets. The shaded area represents the standard deviation.

findings suggest that setting rank = 16 achieves a favorable balance, with the ELoRA parameters accounting for 22.5% and 24.0% of the original model parameters in the respective datasets.

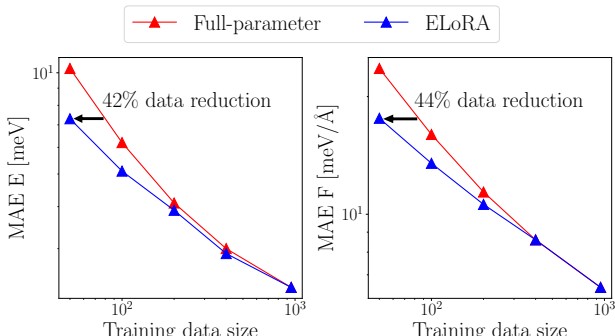

*Figure 4.* **Comparison of energy and force MAE trends between Full-parameter fine-tuning and ELoRA across different training data sizes.** The results are obtained from experiments conducted on the rMD17-aspirin dataset.

### 5.3.2. RANK SETTING

The hyperparameter rank in ELoRA directly influences the number of trainable parameters. To evaluate its impact on model performance, we conduct experiments using rMD17-aspirin and inorganic-Cu datasets. As shown in Figure 5, the results indicate that as rank increases, the validation loss initially decreases but then starts to rise. When the rank is too small, the model tends to underfit the data, whereas a larger rank may lead to overfitting. This aligns with theoretical analysis: as the number of trainable parameters increases, the model transitions from underfitting to overfitting. Specific theoretical analyses can be found in Appendix D. Our

## 6. Related Work

**Neural Network Interatomic Potentials**: The traditional approach to using machine learning for predicting molecular energy and force combines hand-crafted representations of the atomic neighborhood with neural networks (Behler & Parrinello, 2007). Recent advancements have shifted towards end-to-end learnable models based on GNNs, which learn representations directly from atom types and molecular coordinates without requiring hand-crafted features. Early GNN models, such as CGCNN (Xie & Grossman, 2018), DimeNet (Klicpera et al., 2020), SphereNet (Liu et al., 2022), Ewald message passing (Kosmala et al., 2023), PotNet (Lin et al., 2023), and TGT (Hussain et al., 2024) focused on scalar representations that achieve rotational equivariance using invariant features. More recently, equivariant models like PaiNN (Schütt et al., 2021), SEGNN (Brandstetter et al., 2021), NewtonNet (Haghighatlari et al., 2022), and GeoMFormer (Chen et al., 2024) have been proposed, mark-

*Table 4.* **RMSE of 10 different inorganic datasets.** The best results are in bold.

| | Energy RMSE [meV/atom] | | | | | | | Force RMSE [meV/Å] | | | | | | |
|---|---|---|---|---|---|---|---|---|---|---|---|---|---|---|
| | NequIP | Allegro | DPA2 | MACE | MACE | MACE | MACE | NequIP | Allegro | DPA2 | MACE | MACE | MACE | MACE |
| Method | | From scratch | | | Full-parameter | ELoRA | MACE-MP | | From scratch | | | Full-parameter | ELoRA | MACE-MP |
| SSE-PBE | 1.6 | 1.0 | 1.4 | 1.8 | **0.3** | 0.3 | 243.5 | 41.1 | 47.8 | 50.3 | 29.9 | 16.9 | **14.2** | 167.4 |
| H2O-PD | 0.9 | / | **0.5** | 79.9 | 0.6 | 0.8 | 213.1 | 27.1 | / | 24.7 | 29.7 | 19.9 | **16.1** | 1924.9 |
| Ag∪Au-PBE | 42.3 | 39.2 | **2.4** | 369.1 | 10.6 | 8.1 | 534.6 | 43.8 | 58.9 | 17.8 | 34.5 | 11.3 | **9.5** | 530.6 |
| Al∪Mg∪Cu | 38.0 | 18.3 | **2.1** | 7.7 | 3.0 | 2.2 | 919.1 | 48.3 | 40.6 | 19.1 | 42.9 | 9.9 | **8.8** | 227.9 |
| Cu | 6.2 | 1.3 | 1.2 | 38.8 | 0.6 | **0.4** | 381.1 | 16.7 | 8.9 | 8.9 | 13.6 | 5.4 | **4.4** | 190.3 |
| Sn | 18.2 | 5.6 | **4.1** | / | 4.9 | 4.6 | 595.6 | 62.2 | 40.2 | 54.4 | / | 31.7 | **29.2** | 164.9 |
| Ti | 27.6 | 6.9 | **5.0** | 8.3 | 5.9 | 5.8 | 2668.2 | 137.4 | 85.6 | 113.1 | 94.2 | 79.4 | **73.3** | 302.1 |
| V | 8.8 | 4.2 | **4.1** | 14.2 | 4.4 | 4.3 | 128.9 | 91.6 | 82.1 | 90.8 | 140.4 | 74.5 | **68.6** | 318.4 |
| W | 20.8 | **4.0** | 5.6 | 15.6 | 6.1 | 4.7 | 3025.2 | 160.4 | 101.6 | 108.1 | 181.2 | 87.2 | **78.7** | 893.6 |
| HfO2 | 1.5 | 1.4 | 1.0 | 2.3 | 0.5 | **0.3** | 635.8 | 58.8 | 64.0 | 54.2 | **14.7** | 30.0 | 21.0 | 223.8 |

ing a significant advancement in leveraging equivariance for modeling interatomic potentials.

**Parameter-Efficient Fine-Tuning**: Parameter-efficient fine-tuning (PEFT) techniques aim to reduce the number of parameters required for fine-tuning, addressing the inefficiency of fully updating all model parameters as the model size and task complexity grow (Ding et al., 2022). For instance, adapter tuning (Chen et al., 2022) introduces adapter modules with bottleneck architectures between layers. BitFit (Zaken et al., 2022) focuses on updating only the bias terms while keeping the rest of the parameters fixed. LoRA (Hu et al., 2022) reduces the number of trainable parameters by decomposing weight matrices into low-rank matrices. Some theoretical research (Hao et al., 2024; Jang et al., 2024) and variants of LoRA (Berman & Peherstorfer, 2024; Feng et al., 2024; Hayou et al., 2024; Ostapenko et al., 2024; Qin et al., 2024; Zhang & Pilanci, 2024; Zhou et al., 2024) have also been proposed. In the field of GNNs, methods such as GPPT (Sun et al., 2022), MolCPT (Diao et al., 2022), GPF (Fang et al., 2024), and GraphPrompt (Liu et al., 2023) have been introduced. However, these parameter-efficient fine-tuning methods are not applicable to equivariant models, as they will destroy the fundamental equivariance of the model during the fine-tuning process.

## 7. Conclusion

Fine-tuning pre-trained interatomic potentials is crucial to enhancing model accuracy on downstream tasks. Equivariant pre-trained models play a pivotal role in machine learning-based interatomic potentials, maintaining the physical symmetries, which is essential for accurate and reliable atomistic materials simulations. Although various parameter-efficient fine-tuning methods have been made successful in NLP and CV, they fall short when applied to equivariant models because they will not preserve equivariance. To address this challenge, we propose a novel PEFT method called ELoRA. By leveraging path-dependent decomposition for weight updates, ELoRA ensures that the

intrinsic equivariance can be preserved. ELoRA not only guarantees physically consistent predictions, but also enhances data efficiency. Compared to full-parameter fine-tuning, ELoRA achieves SOTA performance on the organic rMD17, 3BPA, and AcAc datasets, as well as on 10 challenging inorganic real cases. ELoRA transforms fine-tuning practices for equivariant models, paving the way for more accurate and efficient atomistic simulations across a wide range of chemical systems.

While our method shows promising results, the following aspects still remain open for further works: (1) Identifying other more effective weight decomposition strategies. (2) Enhancing data efficiency and reducing the number of samples needed for fine-tuning to achieve few-shot or even one-shot learning. (3) Reducing the training epochs required for fine-tuning. (4) Establishing more rigorous proofs for the error generalization bounds of ELoRA.

## Acknowledgments

This work is supported by the following funding: National Science Foundation of China (62032023, 92270206, 62372435, T2125013, 61972377, 61972380, T2293702) and China National Postdoctoral Program for Innovative Talents (BX20240383), and Huawei Technologies Co., Ltd.. The AI-driven experiments, simulations and model training were performed on the robotic AI-Scientist platform of Chinese Academy of Science. We thank Prof. Lin-Wang Wang and Dr. Haibo Li for helpful discussions.

## Impact Statement

This paper presents work whose goal is to advance the field of Machine Learning. There are many potential societal consequences of our work, none which we feel must be specifically highlighted here.

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

## A. Group representations and Equivariance

Group theory provides a formal framework for defining symmetries axiomatically. A representation $D$ of a group $G$ is a function from $G$ to square matrices such that for all $g, h \in G$,

$$D(g)D(h) = D(gh). \tag{9}$$

Formally, a mapping $f : X \to Y$ is equivariant for vector spaces $X$ and $Y$ to a group $G$, if $f \circ D_X(g) = D_Y(g) \circ f$ is satisfied for any $g$ in $G$, where $D_X$ and $D_Y$ represent the transformation of group $G$ on vector spaces $X$ and $Y$, respectively. In this context, we focus on the group of symmetry operations that encompass both isometries of 3D space and permutations of points. When using vectors (e.g., atomic coordinates) in $\mathbb{R}^3$ as input, both $X$ and $Y$ are in $\mathbb{R}^3$, and $f$ can be written as:

$$\vec{y} = f(\vec{x}). \tag{10}$$

We specifically examine the following three types of input transformation and their equivariance properties:

**Permutation**: Let $\mathcal{P}_\sigma$ represents the permutation of vector subscripts, then the permutation equivariance is:

$$f \circ \mathcal{P}_\sigma = \mathcal{P}_\sigma \circ f. \tag{11}$$

**Translation**: Let $\mathcal{T}_{\vec{t}}(\vec{x}) = \vec{x} + \vec{t}$ denote the translation, then the translation equivariance is:

$$f \circ \mathcal{T}_{\vec{t}} = \mathcal{T}_{\vec{t}} \circ f. \tag{12}$$

**Rotation**: $SO(3)$ is the group of 3D rotations. For $g \in SO(3)$, let $\mathcal{R}(g)$ be the representation of $g$ in $\mathbb{R}^3$, then the rotational equivariance is:

$$f \circ \mathcal{R}(g) = \mathcal{R}(g) \circ f. \tag{13}$$

The group elements of $SO(3)$ are represented by $D^l$, called Wigner $D$-matrices (Gilmore, 2008), which map the elements of $SO(3)$ to $(2l + 1) \times (2l + 1)$-dimensional matrices. For scalars and vectors in $\mathbb{R}^3$, the (real) Wigner $D$-matrices are

$$D^0(g) = 1 \qquad \text{and} \qquad D^1(g) = \mathcal{R}(g). \tag{14}$$

In Message Passing Neural Networks (MPNNs), since the ordering of nodes in the graph has no effect on the feature update and the distance between nodes is used for message passing, permutation equivariance and translation equivariance are naturally satisfied. Therefore, EMPNNs mainly focus on how to achieve rotational equivariance.

## B. MACE Architecture

MACE (Batatia et al., 2022b) is an EMPNN model that uses higher body order messages. The expressiveness of the model is improved by using efficient multi-body messages instead of two-body messages. Multi-body messages reduce the number of network layers required to achieve the same expressiveness under two-body messages, resulting in a fast and highly parallelizable model. The MACE architecture follows the general framework of MPNN and includes three parts: message construction, update, and readout.

**Message passing**: In message construction, MACE combines equivariant message passing with efficient many-body messages. The edges are embedded using a learnable radial basis $R_{kl_1l_2l_3}^{(t)}$ and a set of spherical harmonics $Y_{l_1}^{m_1}$, and the self-interaction is performed on the features $h_{j,\tilde{k}l_2m_2}^{(t)}$ with learnable weights $W_{k\tilde{k}l_2}^{(t)}$. $A_{i,kl_3m_3}^{(t)}$ is the two-body feature obtained by pooling neighbor atoms:

$$A_{i,kl_3m_3}^{(t)} = \sum_{l_1m_1,l_2m_2} C_{l_1m_1,l_2m_2}^{l_3m_3} \sum_{j\in\mathcal{N}(i)} R_{kl_1l_2l_3}^{(t)}(r_{ji}) Y_{l_1}^{m_1}(\hat{\boldsymbol{r}}_{ji}) \sum_{\tilde{k}} W_{k\tilde{k}l_2}^{(t)} h_{j,\tilde{k}l_2m_2}^{(t)}, \tag{15}$$

where $C_{l_1m_1,l_2m_2}^{l_3m_3}$ are the standard Clebsch-Gordan coefficients (Griffiths & Schroeter, 2019). The key operation of MACE is to construct a multi-body feature $B_{i,\eta_\nu kLM}^{(t)}$ through the tensor product of the two-body features $A_{i,kl_3m_3}^{(t)}$:

$$B_{i,\eta_\nu kLM}^{(t)} = \sum_{\boldsymbol{lm}} \mathcal{C}_{\eta_\nu,\boldsymbol{lm}}^{LM} \prod_{\xi=1}^{\nu} \sum_{\tilde{k}} w_{k\tilde{k}l_\xi}^{(t)} A_{i,\tilde{k}l_\xi m_\xi}^{(t)}, \tag{16}$$

where the coupling coefficients $\mathcal{C}_{\eta_\nu}^{LM}$ corresponding to the generalised Clebsch-Gordan coefficients and $w_{k\tilde{k}l_\xi}^{(t)}$ is the learnable weight for the self-interaction of $A_{i,kl_3m_3}^{(t)}$. Finally, the message $\boldsymbol{m}_i^{(t)}$ can be written as a linear expansion:

$$m_{i,kLM}^{(t)} = \sum_\nu \sum_{\eta_\nu} W_{z_i kL,\eta_\nu}^{(t)} \boldsymbol{B}_{i,\eta_\nu kLM}^{(t)}. \tag{17}$$

**Update**: The update is a linear function of the message and the residual connection (He et al., 2016):

$$h_{i,kLM}^{(t+1)} = U_t^{kL}(\sigma_i^{(t)}, \boldsymbol{m}_i^{(t)}) = \sum_{\tilde{k}} W_{kL,\tilde{k}}^{(t)} m_{i,\tilde{k}LM} + \sum_{\tilde{k}} W_{z_i kL,\tilde{k}}^{(t)} h_{i,\tilde{k}LM}^{(t)}. \tag{18}$$

**Readout**: The readout is a mapping from the invariant part of the node features to a hierarchical decomposition of site energies:

$$E_i = E_i^{(0)} + E_i^{(1)} + ... + E_i^{(T)}, \qquad \text{where}$$

$$E_i^{(t)} = \mathcal{R}_t\left(\boldsymbol{h}_i^{(t)}\right) = \begin{cases} \sum_{\tilde{k}} W_{\text{readout},\tilde{k}}^{(t)} h_{i,\tilde{k}00}^{(t)} & \text{if } t < T \\ \text{MLP}_{\text{readout}}^{(t)}\left(\left\{h_{i,k00}^{(t)}\right\}_k\right) & \text{if } t = T \end{cases} \tag{19}$$

The readout only depends on the invariant features $h_{i,k00}^{(t)}$, making the site energy contribution $E_i^{(t)}$ invariant.

## C. Proofs

**Proposition C.1.** *The self-interaction of the tensor $h$ is equivalent to the fully connected tensor product of $h$ and $1$.*

*Proof.* Since $1$ is a scalar, $1_{klm} = 1$ when $k = l = m = 0$, and $0$ otherwise.

$$\begin{aligned}
(h \otimes 1)_{k_3 l_3 m_3} &= \sum_{k_1 l_1 m_1, k_2 l_2 m_2} W_{k_3 k_2 k_1, l_3 l_2 l_1} C_{l_1 m_1, l_2 m_2}^{l_3 m_3} h_{k_1 l_1 m_1} 1_{k_2 l_2 m_2} \\
&= \sum_{k_1 l_1 m_1} W_{k_3 k_1, l_3 l_1} C_{l_1 m_1, 00}^{l_3 m_3} h_{k_1 l_1 m_1} 1_{000} \\
&= \sum_{k_1 l_1 m_1} W_{k_3 k_1, l_3 l_1} C_{l_1 m_1, 00}^{l_3 m_3} h_{k_1 l_1 m_1} \\
&= \sum_{k_1} W_{k_3 k_1 l_3} h_{k_1 l_3 m_3}
\end{aligned}$$

In the second equation, the subscripts $k_2$ and $l_2$ in $W$ are omitted because a non-zero result is only possible when $k_2 = l_2 = m_2 = 0$. Similarly, in the last equation, the subscript $l_1$ in $W$ is omitted because $C_{l_1 m_1, 00}^{l_3 m_3} = 1$ when $l_1 = l_3$ and $m_1 = m_3$, and it is zero otherwise. The final form is equivalent to Equation 5, with only a renaming of the subscripts.

**Proposition C.2.** *The residual connection of the tensor $h_i$ is equivalent to the fully connected tensor product of $h_i$ and $\hat{z}_i$, where $\hat{z}_i$ is the one-hot encoding of the chemical species $z_i$.*

*Proof.* The tensor $\hat{z}_i$ is the one-hot encoding of the chemical species $z_i$, where $\hat{z}_{i,klm} = 1$ when $k = z_i, l = m = 0$ and $0$ otherwise.

$$\begin{aligned}
(h_i \otimes \hat{z}_i)_{k_3 l_3 m_3} &= \sum_{k_1 l_1 m_1, k_2 l_2 m_2} W_{k_3 k_2 k_1, l_3 l_2 l_1} C_{l_1 m_1, l_2 m_2}^{l_3 m_3} h_{i,k_1 l_1 m_1} \hat{z}_{i,k_2 l_2 m_2} \\
&= \sum_{k_1 l_1 m_1} W_{k_3 z_i k_1, l_3 l_1} C_{l_1 m_1, 00}^{l_3 m_3} h_{i,k_1 l_1 m_1} \hat{z}_{i,z_i 00} \\
&= \sum_{k_1 l_1 m_1} W_{k_3 z_i k_1, l_3 l_1} C_{l_1 m_1, 00}^{l_3 m_3} h_{i,k_1 l_1 m_1} \\
&= \sum_{k_1} W_{k_3 z_i k_1, l_3} h_{i,k_1 l_3 m_3}
\end{aligned}$$

In the second equation, the subscript $l_2$ in $W$ is omitted because a non-zero result is only possible when $k_2 = z_i$ and $l_2 = m_2 = 0$. Similarly, in the last equation, the subscript $l_1$ in $W$ is omitted because $C_{l_1 m_1, 00}^{l_3 m_3} = 1$ when $l_1 = l_3$ and $m_1 = m_3$, and it is zero otherwise. The final form is equivalent to Equation 6, with only renaming and rearranging of the subscripts.

**Proposition C.3.** *For the pre-trained weight matrix $W^0$ of the fully connected tensor product and its corresponding ELoRA matrices $A$ and $B$, the fully connected tensor product of the tensors $u$ and $v$ with ELoRA is SO(3) equivariant.*

*Proof.* The Clebsch-Gordan coefficients and Wigner $D$-matrices satisfy the following properties. For any element $g$ in the $SO(3)$ group:

$$\sum_{m_1', m_2'} C_{l_1 m_1', l_2 m_2'}^{l_3 m_3} D_{m_1' m_1}^{l_1}(g) D_{m_2' m_2}^{l_2}(g) = \sum_{m_3'} D_{m_3 m_3'}^{l_3}(g) C_{l_1 m_1, l_2 m_2}^{l_3 m_3'}.$$

Then:

$$\sum_{k_1 l_1 m_1, k_2 l_2 m_2} W_{k_3 k_2 k_1, l_3 l_2 l_1} C_{l_1 m_1, l_2 m_2}^{l_3 m_3} (\sum_{m_1'} D_{m_1 m_1'}^{l_1}(g) u_{k_1 l_1 m_1'}) (\sum_{m_2'} D_{m_2 m_2'}^{l_2}(g) v_{k_2 l_2 m_2'})$$

$$= \sum_{k_1 l_1 m_1', k_2 l_2 m_2'} W_{k_3 k_2 k_1, l_3 l_2 l_1} \sum_{m_3'} D_{m_3 m_3'}^{l_3}(g) C_{l_1 m_1', l_2 m_2'}^{l_3 m_3'} u_{k_1 l_1 m_1'} v_{k_2 l_2 m_2'}$$

$$= \sum_{k_1 l_1 m_1, k_2 l_2 m_2} W_{k_3 k_2 k_1, l_3 l_2 l_1} \sum_{m_3'} D_{m_3 m_3'}^{l_3}(g) C_{l_1 m_1, l_2 m_2}^{l_3 m_3'} u_{k_1 l_1 m_1} v_{k_2 l_2 m_2}$$

$$= \sum_{m_3'} D_{m_3 m_3'}^{l_3}(g) \sum_{k_1 l_1 m_1, k_2 l_2 m_2} W_{k_3 k_2 k_1, l_3 l_2 l_1} C_{l_1 m_1, l_2 m_2}^{l_3 m_3'} u_{k_1 l_1 m_1} v_{k_2 l_2 m_2},$$

where the second equation rewrites $m_1'$ and $m_2'$ as $m_1$ and $m_2$. Therefore, the SO(3) equivariance of the fully connected tensor product holds for any matrix $W$. For the pre-trained model, $W_{k_3 k_2 k_1, l_3 l_2 l_1} = W_{k_3 k_2 k_1, l_3 l_2 l_1}^0$. For the fine-tuned model with ELoRA, $W_{k_3 k_2 k_1, l_3 l_2 l_1} = W_{k_3 k_2 k_1, l_3 l_2 l_1}^0 + (B_{l_3 l_2 l_1} A_{l_3 l_2 l_1})_{k_3 k_2 k_1}$.

**Proposition C.4.** *For the pre-trained weight matrix $W^0$ of the fully connected tensor product and its corresponding ELoRA matrices $A$ and $B$, the fully connected tensor product of the tensors $u$ and $v$ with ELoRA satisfies $(u \otimes v)_{k_3 l_3 m_3} = (u \otimes v)_{k_3 l_3 m_3}^0 + \sum_{l_1, l_2} \sum_r B_{k_3 r, l_3 l_2 l_1} (u_{l_1} \otimes v_{l_2})_{r l_3 m_3}$, where $(u \otimes v)_{k_3 l_3 m_3}^0$ is the fully connected tensor product of tensors $u$ and $v$ using $W^0$ without ELoRA and $(u_{l_1} \otimes v_{l_2})_{r l_3 m_3}$ is the fully connected tensor product of tensors $u_{l_1}$ and $v_{l_2}$ using ELoRA matrix $A$.*

*Proof.*

$$(u \otimes v)_{k_3 l_3 m_3} = \sum_{k_1 l_1 m_1, k_2 l_2 m_2} (W_{k_3 k_2 k_1, l_3 l_2 l_1}^0 + (B_{l_3 l_2 l_1} A_{l_3 l_2 l_1})_{k_3 k_2 k_1}) C_{l_1 m_1, l_2 m_2}^{l_3 m_3} u_{k_1 l_1 m_1} v_{k_2 l_2 m_2}$$

$$= (u \otimes v)_{k_3 l_3 m_3}^0 + \sum_{k_1 l_1 m_1, k_2 l_2 m_2} (B_{l_3 l_2 l_1} A_{l_3 l_2 l_1})_{k_3 k_2 k_1} C_{l_1 m_1, l_2 m_2}^{l_3 m_3} u_{k_1 l_1 m_1} v_{k_2 l_2 m_2}$$

$$= (u \otimes v)_{k_3 l_3 m_3}^0 + \sum_{l_1, l_2} \sum_{k_1 m_1, k_2 m_2} (B_{l_3 l_2 l_1} A_{l_3 l_2 l_1})_{k_3 k_2 k_1} C_{l_1 m_1, l_2 m_2}^{l_3 m_3} u_{k_1 l_1 m_1} v_{k_2 l_2 m_2}$$

$$= (u \otimes v)_{k_3 l_3 m_3}^0 + \sum_{l_1, l_2} \sum_{k_1 m_1, k_2 m_2} \sum_r B_{k_3 r, l_3 l_2 l_1} A_{r k_2 k_1, l_3 l_2 l_1} C_{l_1 m_1, l_2 m_2}^{l_3 m_3} u_{k_1 l_1 m_1} v_{k_2 l_2 m_2}$$

$$= (u \otimes v)_{k_3 l_3 m_3}^0 + \sum_{l_1, l_2} \sum_r B_{k_3 r, l_3 l_2 l_1} \sum_{k_1 m_1, k_2 m_2} A_{r k_2 k_1, l_3 l_2 l_1} C_{l_1 m_1, l_2 m_2}^{l_3 m_3} u_{k_1 l_1 m_1} v_{k_2 l_2 m_2}$$

$$= (u \otimes v)_{k_3 l_3 m_3}^0 + \sum_{l_1, l_2} \sum_r B_{k_3 r, l_3 l_2 l_1} (u_{l_1} \otimes v_{l_2})_{r l_3 m_3}$$

**Corollary C.5.** *For the pre-trained weight matrix $W^0$ of the fully connected tensor product and its corresponding ELoRA matrices $A$ and $B$, $A_{l_3 l_2 l_1}$ projects the equivariant message on the path $(l_1, l_2, l_3)$ in the fully connected tensor product of the tensors $u$ and $v$ with ELoRA into a low-dimensional space of dimension R.*

# D. Generalization Theory

The PAC-Bayesian framework (Shawe-Taylor & Williamson, 1997; McAllester, 1998; 1999) is an analytical approach that combines Probably Approximately Correct (PAC) theory with Bayesian theory to study the generalization performance of randomized learning algorithms. By introducing Bayesian prior and posterior distributions into the analysis of generalization error in PAC theory, it establishes generalization bounds for arbitrary priors and provides probabilistic guarantees for the generalization error.

Let $\mathbf{w}$ represent the parameters of the neural network, and let $f_{\mathbf{w}}(x) : \mathcal{X} \to \mathcal{Y}$ denote the mapping from input $x$ to output $y$ as computed by the neural network. Assume that the data points $(x_1, y_1), (x_2, y_2), \ldots, (x_N, y_N)$ are independently drawn from a distribution $\mathcal{D}$. Define $\ell$ as the loss function, where the loss on a sample $(x, y)$ is given by $\ell(f_{\mathbf{w}}(x), y)$. The generalization error of the neural network on $\mathcal{D}$ is expressed as $L(f_{\mathbf{w}}) = E_{x,y \sim \mathcal{D}}[\ell(f_{\mathbf{w}}(x), y)]$, while the empirical error on a dataset is $\hat{L}(f_{\mathbf{w}}) = \frac{1}{N} \sum_{i=1}^{N} [\ell(f_{\mathbf{w}}(x_i), y_i)]$.

Since the underlying distribution $\mathcal{D}$ is typically unknown, the generalization error can only be approximated through the empirical error computed on a test set. The PAC-Bayesian framework provides an estimation of the generalization bound $L(f_{\mathbf{w}}) - \hat{L}(f_{\mathbf{w}})$ for randomized predictors. Recent studies based on PAC-Bayesian theory have derived generalization bounds for models such as Multi-Layer Perceptrons (MLPs) (Neyshabur et al., 2018) and Message Passing Neural Networks (MPNNs) (Ju et al., 2023). These bounds are typically derived using the spectral norm of weight matrices, which is often proportional to the stable rank $\frac{\|W\|_F^2}{\|W\|_2^2}$. Below, we present the propositions and their corollaries for MLPs and MPNNs.

**Lemma D.1.** *(Neyshabur et al., 2018) For any $B, d, h > 0$, let $f_{\mathbf{w}} : \mathcal{X}_{B,n} \to \mathbb{R}^k$ be a $d$-layer feedforward network with ReLU activations. Then, for any $\delta, \gamma > 0$, with probability $\geq 1 - \delta$ over a training set of size $m$, for any $\mathbf{w}$, we have:*

$$L_0(f_{\mathbf{w}}) \leq \hat{L}_{\gamma}(f_{\mathbf{w}}) + \mathcal{O}\left( \sqrt{\frac{B^2 d^2 h \ln(dh) \Pi_{i=1}^{d} \|W_i\|_2^2 \sum_{i=1}^{d} \frac{\|W_i\|_F^2}{\|W_i\|_2^2} + \ln \frac{dm}{\delta}}{\gamma^2 m}} \right).$$

**Corollary D.2.** *For a $d$-layer feed-forward network, the generalization bound is related to the stable rank of the weight matrix as follows:*

$$\mathcal{O}\left( \sqrt{\frac{\Pi_{i=1}^{d} \|W_i\|_2^2 \sum_{i=1}^{d} \frac{\|W_i\|_F^2}{\|W_i\|_2^2}}{N}} \right),$$

*where $N$ is the number of samples.*

**Lemma D.3.** *(Ju et al., 2023) Suppose all of the nonlinear activations in $\{\phi_t, \rho_t, \psi_t : \forall t\}$ and the loss function $\ell(\cdot, y)$ (for any fixed label $y \in \mathcal{Y}$) are twice-differentiable, Lipschitz-continuous and their first-order and second-order derivatives are both Lipschitz-continuous. With probability at least $1 - \delta$ over the randomness of $N$ independent samples from $\mathcal{D}$, for any $\delta > 0$, and any $\epsilon > 0$ close to zero, any model $f$ with weight matrices in the set $\mathcal{H}$ satisfies:*

$$L(f) \leq (1 + \epsilon)\hat{L}(f) + O\left( \frac{\log(\delta^{-1})}{N^{3/4}} \right) + \sum_{i=1}^{l} \sqrt{\frac{CBd_i \left( \max_{(X,G,y) \sim \mathcal{D}} \|X\|^2 \|P_G\|^{2(l-1)} \right) \left( r_i^2 \prod_{j=1}^{l} s_j^2 \right)}{N}},$$

*where $B$ is an upper bound on the value of the loss function $\ell(x, y)$ for any $(x, y) \sim \mathcal{D}$, $C$ is a fixed constant depending on the activation, and the loss function's Lipschitz-continuity.*

**Corollary D.4.** *For a $l$-layer message-passing neural network, the generalization bound is related to the stable rank of the weight matrix as follows:*

$$\mathcal{O}\left( \sum_{i=1}^{l} \sqrt{\frac{\|W_i\|_2^2 \Pi_{j=1}^{l} \frac{\|W_j\|_F^2}{\|W_j\|_2^2}}{N}} \right),$$

*where $N$ is the number of samples.*

Corollaries D.2 and D.4 indicate that reducing the stable rank of the weight matrix leads to a tighter generalization bound. Therefore, we focus on the upper bound of the stable rank. The upper bound of the stable rank of the fine-tuned weight matrix can be determined using the following propositions.

**Lemma D.5.** *For any matrix $A \in \mathbb{R}^{n \times m}$, its stable rank is defined as $srank(A) = \frac{\|A\|_F^2}{\|A\|_2^2}$, where $\|A\|_F$ is the Frobenius norm of $A$ and $\|A\|_2$ is the spectral norm of $A$, then $srank(A) \leq rank(A)$.*

*Proof.* Let $\sigma_1, \sigma_2, \ldots, \sigma_r$ be the non-zero singular values of $A$ from large to small, then $r = \text{rank}(A)$ is the rank of $A$, $\|A\|_F = \sqrt{\sum_{i=1}^r \sigma_i^2}$, $\|A\|_2 = \sigma_1$. The definition of stable rank can be written as $\text{srank}(A) = \frac{\sum_{i=1}^r \sigma_i^2}{\sigma_1^2}$. Note that $\sigma_1^2 \geq \sigma_i^2$ holds for all $i = 1, 2, \ldots, r$, so $\sum_{i=1}^r \sigma_i^2 \leq r \cdot \sigma_1^2$. Then $\text{srank}(A) = \frac{\sum_{i=1}^r \sigma_i^2}{\sigma_1^2} \leq r = \text{rank}(A)$.

**Proposition D.6.** *For the pre-trained weight matrix $W^0 \in \mathbb{R}^{d \times k}$ and its update $\Delta W \in \mathbb{R}^{d \times k}$, $srank(W^0 + \Delta W) \leq rank(W^0) + rank(\Delta W)$.*

*Proof.* $\text{srank}(W^0 + \Delta W) \leq \text{rank}(W^0 + \Delta W) \leq \text{rank}(W^0) + \text{rank}(\Delta W)$.

**Proposition D.7.** *For the pre-trained weight matrix $W^0 \in \mathbb{R}^{d \times k}$ and its update $\Delta W \in \mathbb{R}^{d \times k}$, if $\epsilon = \frac{\|\Delta W\|_2}{\|W^0\|_2} < 1$, then $\sqrt{srank(W^0 + \Delta W)} \leq \frac{1}{1-\epsilon}(\sqrt{srank(W^0)} + \epsilon \cdot \sqrt{srank(\Delta W)})$.*

*Proof.*

$$
\begin{aligned}
\sqrt{\text{srank}(W^0 + \Delta W)} &= \frac{\|W^0 + \Delta W\|_F}{\|W^0 + \Delta W\|_2} \\
&\leq \frac{\|W^0\|_F + \|\Delta W\|_F}{\|W^0\|_2 - \|\Delta W\|_2} \\
&= \frac{\|W^0\|_2}{\|W^0\|_2 - \|\Delta W\|_2}\left(\frac{\|W^0\|_F}{\|W^0\|_2} + \frac{\|\Delta W\|_F}{\|\Delta W\|_2}\frac{\|\Delta W\|_2}{\|W^0\|_2}\right) \\
&= \frac{1}{1-\epsilon}(\sqrt{\text{srank}(W^0)} + \epsilon \cdot \sqrt{\text{srank}(\Delta W)})
\end{aligned}
$$

From Propositions D.6 and D.7, we observe that the upper bound of the stable rank of the fine-tuned weight matrix is determined by the rank or stable rank of $\Delta W$. Consequently, LoRA can achieve a tighter generalization bound compared to full-parameter fine-tuning by reducing the rank of $\Delta W$. When $\Delta W$ has a low rank, the empirical error of the LoRA fine-tuned model is relatively high, resulting in underfitting. As the rank of $\Delta W$ increases, the empirical error decreases, but the generalization performance worsens, causing the model to transition from underfitting to overfitting. At the balance point, where the empirical errors of full-parameter fine-tuning and LoRA are equal, LoRA is expected to exhibit superior generalization performance. This process is consistent with the description in Section 5.3.2.

Furthermore, Corollary D.4 shows that the generalization bound is inversely proportional to the number of samples. This implies that as the number of samples increases, the difference between LoRA and full-parameter fine-tuning gradually diminishes, aligning with the observations described in Section 5.3.1.

# E. Experimental Details

## E.1. Pre-trained Datasets

**SPICE** (Eastman et al., 2023): SPICE contains small molecules of up to 50 atoms and involves ten chemical elements: H, C, N, O, F, P, S, Cl, Br, and I. To facilitate the learning of intramolecular non-bonded interactions, it also uses a few larger molecules of 50-90 atoms randomly selected from the QMugs dataset (Isert et al., 2022).

**MPTrj** (Deng et al., 2023): MPTrj consists of a large collection of static calculations and structural optimization trajectories from the Materials Project (MP) (Jain et al., 2013), including approximately 1.5M configurations of 150k unique inorganic crystal structures.

## E.2. Downstream Datasets

**rMD17** (Christensen & Von Lilienfeld, 2020): The revised MD17 (rMD17) is a recomputed version of the original MD17 dataset (Chmiela et al., 2017) at higher numerical accuracy. This dataset contains long molecular dynamics trajectories of

ten small organic molecules, with 100,000 structures each. It recomputes energies and forces at the PBE/def2-SVP level of theory using very tight SCF convergence and a very dense DFT integration grid to reduce numerical noise.

**3BPA** (Kovács et al., 2021): The 3BPA dataset is sampled from molecular dynamics simulation of the large and flexible drug-like molecule 3-(benzyloxy)pyridin-2-amine. It is used to test the extrapolation ability of a model. Its training set contains 500 structures sampled at 300 K, and three test sets contain structures sampled at 300 K, 600 K, and 1200 K to evaluate the accuracy both in and out of domain. The fourth test set consists of optimized dihedral slices by rotating the dihedral angles to produce PES regions far away from the training data.

**AcAc** (Batatia et al., 2022a): The AcAc dataset is a dataset for evaluating the extrapolation ability of the model to higher temperatures, bond breaking, and bond torsion for the acetylacetone molecule. The training set is sampled at 300 K, and the test set is sampled independently at 300 K and 600 K.

**Ag∪Au-PBE** (Wang et al., 2021), **Al∪Mg∪Cu** (Jiang et al., 2021): The Ag∪Au-PBE dataset consists of Ag and Au configurations generated using the DP-GEN scheme, while the Al∪Mg∪Cu dataset includes Al, Mg, and Cu configurations produced by the same method. All DFT calculations are performed with the VASP (Kresse & Furthmüller, 1996a;b) software, applying the PBE exchange-correlation functional.

**H2O-PD** (Zhang et al., 2021): The water/ice dataset is used to train the model to calculate the water phase diagram. This dataset is labeled by the VASP software using the SCAN exchange-correlation functional.

**SSE-PBE** (Huang et al., 2021): This dataset contains the solid electrolytes $Li_{10}GeP_2S_{12}$ and $Li_{10}SiP_2S_{12}$ generated by the DP-GEN method. All DFT calculations are performed with the VASP software, applying the PBE exchange-correlation functional.

**Other datasets**: The Cu (Zhang et al., 2020), Sn (Chen et al., 2023), Ti (Wen et al., 2021), V (Wang et al., 2022a), W (Wang et al., 2022b), and HfO2 (Wu et al., 2021) datasets are generated by the DP-GEN method under complex temperature and pressure conditions.

We list the sizes and links of these datasets in Table 5.

*Table 5.* **Sizes and links of different datasets.**

| Dataset | Size | Link |
| --- | --- | --- |
| rMD17 | 1,000,000 | https://dx.doi.org/10.6084/m9.figshare.12672038 |
| 3BPA | 13,997 | https://github.com/davkovacs/BOTNet-datasets |
| AcAc | 6,263 | https://github.com/davkovacs/BOTNet-datasets |
| SSE-PBE | 15,774 | https://www.aissquare.com/datasets/detail?pageType=datasets&id=146 |
| H2O-PD | 48,419 | https://www.aissquare.com/datasets/detail?pageType=datasets&id=137 |
| Ag∪Au-PBE | 17,508 | https://www.aissquare.com/datasets/detail?pageType=datasets&id=152 |
| Al∪Mg∪Cu | 25,397 | https://www.aissquare.com/datasets/detail?pageType=datasets&id=139 |
| Cu | 15,366 | https://www.aissquare.com/datasets/detail?pageType=datasets&id=132 |
| Sn | 6,725 | https://www.aissquare.com/datasets/detail?pageType=datasets&id=129 |
| Ti | 10,528 | https://www.aissquare.com/datasets/detail?pageType=datasets&id=133 |
| V | 15,673 | https://www.aissquare.com/datasets/detail?pageType=datasets&id=135 |
| W | 44,397 | https://www.aissquare.com/datasets/detail?pageType=datasets&id=136 |
| HfO2 | 28,577 | https://www.aissquare.com/datasets/detail?pageType=datasets&id=145 |

### E.3. Models

**ACE** (Kovács et al., 2021): The Atomic Cluster Expansion (ACE) is a framework that linearly parameterizes potential energy surfaces using body-ordered symmetric polynomials. Its key innovation is the density trick, which projects atomic neighborhoods onto invariant basis functions, allowing efficient computation of high-order interactions with costs scaling linearly with neighboring atoms, combining accuracy with efficiency.

**NequIP** (Batzner et al., 2022): NequIP is an E(3)-equivariant graph neural network that leverages geometric tensor representations, including scalars, vectors, and higher-order tensors, to model interatomic potentials. Its innovative use of equivariant convolutions and tensor features ensures that internal operations preserve symmetry transformations, enabling more accurate and data-efficient representations of atomic environments.

**BOTNet** (Batatia et al., 2022a): BOTNet (Body Ordered Equivariant Network) simplifies NequIP's design by adopting a body-ordered structure, incrementally building interaction complexity through layers. It removes non-essential nonlinearities while retaining NequIP's equivariant framework, achieving similar accuracy with improved interpretability and efficiency.

**Allegro** (Musaelian et al., 2023): Allegro is an innovative deep learning architecture for molecular dynamics, which combines strict locality and equivariance in its design. Unlike message-passing neural networks, it employs tensor product layers to directly model many-body interactions without iterative propagation, enabling efficient parallel computation and scalability to large systems.

**DPA2** (Zhang et al., 2023): DPA-2 is an advanced large atomic model (LAM) that leverages a multi-task pre-training framework to embed diverse chemical and configurational knowledge. It employs fine-tuning to adapt the pre-trained model for specific downstream tasks, achieving high generalizability and efficiency across a wide range of molecular and material systems.

**PACE** (Xu et al., 2024): PACE approximates a broader class of $SE(3) \times S_n$ equivariant polynomial functions with higher degrees by combining spherical harmonics, tensor products, and an edge booster. It extends ACE with symmetric contractions to capture many-body interactions, achieving strong accuracy and generalization in force field prediction.

### E.4. Training Details

To facilitate reproducibility, Table 6 summarizes the training hyperparameters for different MACE models. The settings for MACE-From-scratch (Batatia et al., 2022b), MACE-MP (Batatia et al., 2023), and MACE-OFF (Kovács et al., 2023) are derived from previously published studies, while those for MACE-MP-Fine-tune and MACE-OFF-Fine-tune reflect the fine-tuning hyperparameters used in this work. The MACE model code is modified from the main branch of the open source GitHub repository at https://github.com/ACEsuit/mace (commit hash: 346a829f).

*Table 6.* **Training hyperparameters for different MACE models.**

|  | MACE-From-scratch | MACE-MP | MACE-OFF | MACE-MP-Fine-tune | MACE-OFF-Fine-tune |
|---|---|---|---|---|---|
| model | ScaleShiftMACE | ScaleShiftMACE | MACE | ScaleShiftMACE | MACE |
| rmax | 5.0 | 6.0 | 5.0 | 6.0 | 5.0 |
| num_radial_basis | 8 | 10 | 8 | 10 | 8 |
| num_channels | 256 | 128 | 128 | 128 | 128 |
| max_L | 2 | 2 | 1 | 2 | 1 |
| loss | ef | universal | ef | ef | ef |
| forces_weight | 1000 | 10 | 1000 | 1000 | 1000 |
| energy_weight | 1 | 1 | 40 | 1 | 1 |
| lr | 0.01 | 0.005 | 0.01 | 0.005 | 0.005 |
| weight_decay | 5e-7 | 1e-8 | 5e-10 | 1e-8 | 1e-8 |
| scheduler_patience | 50 | 5 | 20 | 5 | 5 |
| ema_decay | 0.99 | 0.995 | 0.99 | 0.995 | 0.995 |
| clip_grad | 10 | 100 | 1 | 100 | 100 |

## F. Other Experimental Results

### F.1. Comparison of Different Fine-tuning Methods

We compare two other popular fine-tuning methods: Readout fine-tuning (freezing the previous layers and only fine-tuning the Readout layers) and Adapter fine-tuning. Table 7 depicts the RMSE results of these fine-tuning methods on Cu and Sn datasets. Adapter fine-tuning performs worse because the equivariance is destroyed under the fine-tuning process. Readout fine-tuning is not as accurate as full-parameter fine-tuning because only the last few layers are fine-tuned, which lacks flexibility compared to full-parameter fine-tuning.

We also conduct experiments with ELoRA on SevenNet (Park et al., 2024), which is another representative SO(3)-equivariant MPNN. As shown in Table 8, the results are consistent with those in Table 7.

*Table 7.* **RMSE of energy (E, meV/atom) and force (F, meV/Å) for different finetuning methods on MACE.** The best results are in bold.

| Method | | MACE Full-parameter | MACE Adapter | MACE Readout | MACE ELoRA |
|---|---|---|---|---|---|
| Cu | E | 0.6 | 4.4 | 5.6 | **0.4** |
| | F | 5.4 | 22.9 | 28.2 | **4.4** |
| Sn | E | 4.9 | 27.6 | 30.8 | **4.6** |
| | F | 31.7 | 67.5 | 74.8 | **29.2** |
| Trainable parameters | | 723,866 | 169,296 | 2,192 | 176,666 |

*Table 8.* **RMSE of energy (E, meV/atom) and force (F, meV/Å) for different finetuning methods on SevenNet.** The best results are in bold.

| Method | | SevenNet Full-parameter | SevenNet Adapter | SevenNet Readout | SevenNet ELoRA |
|---|---|---|---|---|---|
| Cu | E | 0.9 | 2.5 | 10.2 | **0.8** |
| | F | 12.8 | 32.1 | 153.2 | **12.2** |
| Sn | E | 3.4 | 6.8 | 16.4 | **3.0** |
| | F | 74.1 | 117.0 | 190.5 | **73.4** |

## F.2. Stable Rank

In Appendix D, we analyze how low-rank adaptation reduces the stable rank of the update to the weight matrix, $\Delta W$, during fine-tuning. Using the rMD17-aspirin dataset as an example, we assess the stable rank of $\Delta W$ for each layer of the model under both full-parameter fine-tuning and ELoRA fine-tuning. The results, shown in Figure 6, indicate that ELoRA fine-tuning generally leads to a lower stable rank for $\Delta W$.

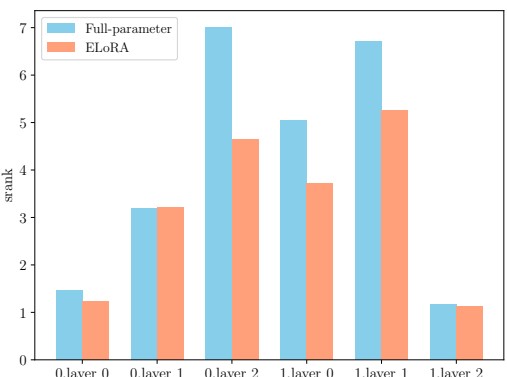

*Figure 6.* **Comparison of stable rank for $\Delta W$ between full-parameter fine-tuning and ELoRA fine-tuning across different layers.** Results are from models trained on the rMD17-aspirin dataset.

## F.3. Hyperparameters

Since full-parameter fine-tuning and ELoRA update different numbers of parameters, they may require different learning rates to achieve optimal performance. We test various learning rates, and the results are shown in the left panel of Figure 7. Overall, ELoRA outperforms full-parameter fine-tuning, with the best ELoRA model surpassing the best full-parameter fine-tuning model. This eliminates the potential influence of learning rate settings on the performance comparison between

the two approaches.

ELoRA improves generalization by reducing the number of parameters being trained, which helps prevent overfitting. We aim to test whether other regularization methods could achieve similar effects. Typically, weight decay and dropout are used for regularization. However, in SO(3) equivariant GNNs, applying dropout directly could disrupt the network's equivariance. Therefore, we only test weight decay, and the results of using different weight decay hyperparameters in full-parameter fine-tuning are shown in the right panel of Figure 7. As we can see, weight decay do not provide significant benefits in this case.

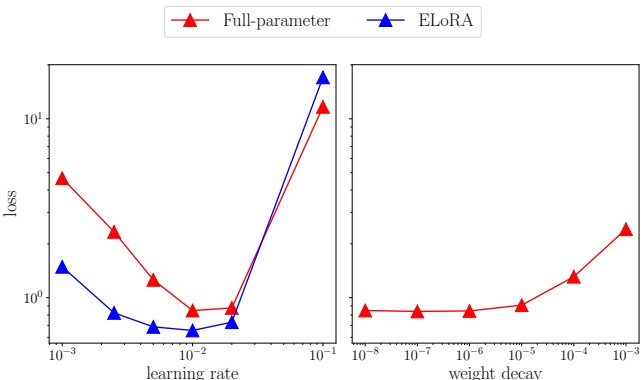

*Figure 7.* **The impact of learning rates and weight decay on model loss.** Results are from models trained on the rMD17-aspirin dataset.

