# OpenReview forum: "ELoRA: Low-Rank Adaptation for Equivariant GNNs"
_ICML.cc/2025/Conference — ICML 2025 poster_

### Official Review · Reviewer_G2Qy · 2025-02-23

**Overall Recommendation:** 3

**Summary:**

The paper introduces Equivariant Low-Rank Adaptation, a parameter-efficient fine-tuning method for pre-trained equivariant Graph Neural Networks used in interatomic potential modeling. Unlike existing fine-tuning approaches that break equivariance, ELoRA employs a path-dependent low-rank decomposition to update weights while preserving equivariance, ensuring physically consistent predictions. Theoretical proofs confirm its equivariance preservation, and experiments on organic and inorganic datasets demonstrate that ELoRA significantly improves energy and force prediction accuracy over full-parameter fine-tuning, reducing data requirements while maintaining efficiency.

**Claims And Evidence:**

The claims made in the submission are supported by clear and convincing evidenc

**Essential References Not Discussed:**

NA

**Experimental Designs Or Analyses:**

The experiments are reasonable and sound.

**Methods And Evaluation Criteria:**

Yes

**Other Comments Or Suggestions:**

See Strengths And Weaknesses

**Other Strengths And Weaknesses:**

Strengths:
1. Preserves Equivariance in Fine-Tuning – Unlike traditional fine-tuning approaches that break equivariance, ELoRA ensures that equivariance is maintained throughout the adaptation process, which is crucial for physically consistent predictions in interatomic potential modeling.
2. Improves Data Efficiency – By leveraging low-rank adaptations, ELoRA significantly reduces the amount of training data required for fine-tuning while maintaining or even improving predictive accuracy, making it highly practical for resource-intensive scientific applications.
3. Strong Theoretical Foundation – The paper provides rigorous mathematical proofs demonstrating that ELoRA preserves equivariance and effectively projects equivariant messages into a lower-dimensional space.
4. Superior Performance on Benchmarks – Experimental results on organic (rMD17) and inorganic datasets show that ELoRA achieves state-of-the-art accuracy, outperforming both full-parameter fine-tuning and models trained from scratch in energy and force prediction.

Weaknesses & Questions:
For Figure 3 and Eq. 8:
(1) What does $K$ mean? Does it mean the number of channels? And $K_0^1$ means the number of channels of the first tensor when $l=0$?
 (2) The superscript of K is the index instead of the power, right? The notation makes me confused. In Eq.7, the $k$ uses subscript, but in Fig. 3, the $K$ uses superscript.
(3) I think the notation × has different meanings. In line 254, it is used to denote the dimension of a matrix. But in other places, does it mean product?  For example, in line 256-257, do you mean $R \ll \min (K_{l_3}^3, K_{l_2}^2 \cdot K_{l_1}^1 )$?
 (4) How to get $K^3$? Is it a hyperparameter or is it decided by $K^1$ and $K^2$?

For the method and setting:
(5) So this papers assumes $R \ll \min (K_{l_3}^3, K_{l_2}^2 \times K_{l_1}^1 )$. But in some real-world scenarios, the number of channels = 1. For example, many datasets only have the atomic number when l=0 and the atom position when l=1. In this case, is Elora still meaningful? Could the authors provide more information about the datasets used? For example, the datasets in E.1 and E.2, how many channels do they have? And could you provide the table of the number of parameters to tune with and without ELoRA?

For the implementation:
(6) No code available.
(7) Did you implement ELoRA based on e3nn?

**Questions For Authors:**

See Weaknesses & Questions

**Relation To Broader Scientific Literature:**

NA

**Theoretical Claims:**

I checked the proofs. The most important one is the proof of equivariance of ELora, it's correct.

---

> ### Author Rebuttal · Authors · 2025-04-01
>
> Thank you for the valuable comments and suggestions.
>
> **Q1: The meaning of $K$.**
>
> A1: $K$ means the number of channels. $K_0^1$ means the number of channels of the first tensor when $l=0$.
>
> **Q2: The meaning of the superscript of $K$.**
>
> A2: The superscript of $K$ does not represent a power. The $K$ in Figure 3 corresponds to the $K$ in the equations of Section 4.2. Here, the superscript of $K$ is used to identify different tensors, while in Equation (7), the subscript of $k$ is used for the same purpose. We will address this in our revised paper.
>
> **Q3: The ambiguity of the multiplication symbol.**
>
> A3: In line 254, it is used to denote the dimension of a matrix. In lines 256–257, it means $R \ll \min(K^3_{l_3}, K^2_{l_2} \cdot K^1_{l_1})$. The use of the multiplication symbol here might cause misunderstanding, and we will revise it.
>
> **Q4: The choice of $K_3$.**
>
> A4: $K_3$ is a hyperparameter of the neural network and can be specified arbitrarily.
>
> **Q5: The connection between $K$ and datasets.**
>
> A5: Here, $K_1$, $K_2$, and $K_3$ are not uniquely determined by the dataset; rather, they are hyperparameters of the network and can be set arbitrarily. This is similar to the hidden dimensions in an MLP, which can be larger than the input dimension. The number of channels used in our model is provided in Section E.4, which is typically set to 128.
>
> **Q6: Code availability.**
>
> A6: ELoRA is implemented based on e3nn. We have provided the code in the anonymous repository https://anonymous.4open.science/r/ELoRA/README.md.

---

> > ### Comment · Reviewer_G2Qy · 2025-04-04
> >
> > Thanks for your responses.
> > 1. Would you mind providing more information about Q5? There are many questions in Q5, they are not fully addressed.
> > 2. In the code link you provided, I didn't find the implementation of the model. Could you point it to me or update the README?

---

> > > ### Author Response · Authors · 2025-04-04
> > >
> > > Thanks. We apologize for not addressing all of your concerns in our previous response. We will provide more detailed explanations and clarifications to address your concerns better.
> > >
> > > **Q1: The connection between $K$ and datasets.**
> > >
> > > A1: $K^1$, $K^2$, and $K^3$ are not uniquely determined by the dataset; rather, they are hyperparameters of the network and can be set arbitrarily. Therefore, although the input includes only atomic number and atomic position, the node features within the model are of higher rotation orders and have multiple channels.
> > >
> > > In the paper, we assume that $R \ll \min(K^3_{l_3}, K^2_{l_2} \times K^1_{l_1})$, where $R$ is a hyperparameter. The input of the dataset in E.1 and E.2 includes atomic number ($l=0$) and atomic position ($l=1$), both with 1 channel. In the model, atomic number is one-hot encoded, so the number of channels for $l=0$ increases from 1 to the number of element types. Then, a self-interaction operation (Section 4.1, Equation (5): $\sum_{\tilde{k}} W_{k\tilde{k} l} h_{\tilde{k} l m}$) is used to project the channel dimension to a higher dimension, which is typically set to 128.
> > >
> > > In the interaction block, the point convolution is applied to obtain new node features, as described by Equation (3) in Section 4.1:
> > > $$
> > > \sum_{l_1 m_1, l_2 m_2} R_{k l_1 l_2 l_3}(r_{ji})Y_{m_1}^{l_1}(\vec{r_{ji}})  \otimes h_{j, k l_2 m_2},
> > > $$
> > > Where $Y_{m_1}^{l_1}(\vec{r})$ is the spherical harmonic, and its rotation order $l$ is a hyperparameter that can be freely specified. In the paper, it is set to $l=0,1,2$. After the first interaction block, the node features include components of rotation order $l=0$, $l=1$, and $l=2$, each with 128 channels.
> > >
> > > For the datasets in Sections E.1 and E.2, their inputs consist of atomic number and position, and the node features typically have 128 channels and a maximum rotation order of $l=2$. Therefore, $K$ can reach 128 in the model for each dataset. In this case, using ELoRA is meaningful.
> > >
> > > **Q2: The number of parameters to tune with and without ELoRA.**
> > >
> > > ​A2: We list the number of trainable parameters required by different fine-tuning methods in response to Reviewer S1HM’s Q2, and also compare two other popular fine-tuning methods: Readout fine-tuning (freezing the previous layers and only fine-tuning the Readout layers) and Adapter fine-tuning. We present the results again here.
> > >
> > > **Table 1 The comparasion of four different finetuning methods.**
> > >
> > > ||MACE (Full-parameter)|MACE (Adapter)|MACE (Readout)|MACE (ELoRA)|
> > > |:-:|:-:|:-:|:-:|:-:|
> > > |Cu,E|0.6|4.4|5.6|**0.4**|
> > > |Cu,F|5.4|22.9|28.2|**4.4**|
> > > |Sn,E|4.9|27.6|30.8|**4.6**|
> > > |Sn,F|31.7|67.5|74.8|**29.2**|
> > > |Number of trainable parameters|723866|169296|2192|176666|
> > >
> > > **Q3: Code availability.**
> > >
> > > A3: The model we use is based on the open-source model MACE (https://github.com/ACEsuit/mace, commit hash: 346a829f).
> > >
> > > The anonymous repository we provide (https://anonymous.4open.science/r/ELoRA/README.md) contains the implementation of ELoRA based on the e3nn library. We add parameters for fine-tuning to nn.FullyConnectedNet, o3.TensorProduct, and o3.Linear. The main modifications can be found at the following locations:
> > >
> > > - https://anonymous.4open.science/r/ELoRA/e3nn/nn/_fc.py, lines 20–26,
> > > - https://anonymous.4open.science/r/ELoRA/e3nn/o3/_tensor_product/_tensor_product.py, lines 391–408,
> > > - https://anonymous.4open.science/r/ELoRA/e3nn/o3/_linear.py, lines 227–236.
> > >
> > > To use ELoRA, one needs to install the modified version of e3nn from our anonymous repository (https://anonymous.4open.science/r/ELoRA) to replace the original e3nn library.

---

### Official Review · Reviewer_S1HM · 2025-03-04

**Overall Recommendation:** 3

**Summary:**

The paper presents *ELoRA*, a novel method for fine-tuning equivariant GNNs that preserves the essential equivariance property, addressing limitations of traditional fine-tuning approaches. ELoRA demonstrates significant improvements in model performance. The method employs a path-dependent weight update decomposition strategy and low-rank adaptation to enhance data efficiency while maintaining physical consistency, thereby advancing the understanding of pretraining-fine-tuning paradigms in the context of materials science and chemical simulations.

**Claims And Evidence:**

Some of the paper’s claims have minor issues. A few statements are not well-supported, or require small changes to be made correct.

**Essential References Not Discussed:**

No.

**Experimental Designs Or Analyses:**

- For the comparison of fine-tuning methods, the authors only compared ELoRA and FFT, and other parameter fine-tuning methods should be added for comparison.
- In Table 2, MACE-ELoRA performed best on only 3 datasets (under the same architecture) for Energy RMSE. The authors should add comparisons with fine-tuning methods such as ELoRA, downstream fine-tune in other architectures.
- With regard to equivariance, the authors should add case studies to illustrate the effectiveness of the proposed method.

**Methods And Evaluation Criteria:**

I believe that the proposed methods and evaluation criteria make sense for the problem or application at hand. However, I think the author needs to give examples to show that the proposed method does maintain equivariance.

**Other Comments Or Suggestions:**

- Reference error: GNNadapter in L69 should be AdapterGNN.
- Figure 1 should give more captions for understanding, e.g. dark blue/light blue circles, the meaning of the dotted box.
- For propositions with proof, a reference link can be provided.

**Other Strengths And Weaknesses:**

**Strengths**:
- The contribution is both original and strong.
- Sufficient theoretical proof is provided.
- Experiments have proven their excellent performance.
- The main architecture is clear and provides detailed explanations.

**Weaknesses**:
- This paper is ill-motivated. There is a contradiction: L160 "The pre-trained training datasets at first-principles accuracy are often sparse...", and the diversity of data in the Introduction "due to the diversity of material structures..."
- The readability of the paper is poor, especially in Section 3.
    - What is mean of L150-153: “They can only learn from the data (the dark blue circle in Figure 1(b)) in the specific downstream task they are trained on…”
    - Some terms should be standardized for academic expressions, e.g. complex downstream task data in L158 should refer to OOD data.
    - The organization of the paper is confusing, for example, the method mentioned in Section 3 (active learning/DP-GEN) does not seem to be relevant to this paper, and if it is, it should clarify the relationship with the existing works (limitation, improvement, etc.).
- The comparative experiment is inadequate and unconvincing.
- In general, the author proposes that the method is not technically novel.

**Questions For Authors:**

- Does path-dependent weight increase the number of parameters? Please provide the number of parameters for the different fine-tuning methods.
- please refer to the comments above.

**Relation To Broader Scientific Literature:**

No.

**Theoretical Claims:**

I double-checked the theory as well as the proof of the appendix and found no major errors.

---

> ### Author Rebuttal · Authors · 2025-04-01
>
> Thank you for your feedback and we will correct all the typos, improve the writing and enhance the readability according to your comments. We will polish the describtion of Figure1 and add reference link for the proved propositions.
>
> **Q1: The novelty of ELoRA.**
>
> A1: We propose a PEFT method for SO(3)-equivariant MPNNs, which is technically novel in its path-dependent low-rank adaptation. To the best of our knowledge, no existing PEFT methods preserve equivariance.
>
> **Q2: The comparative experiment of different finetuning methods.**
>
> A2: We will compare two other popular fine-tuning methods: Readout fine-tuning (freezing the previous layers and only fine-tuning the Readout layers) and Adapter fine-tuning. Table 1 depicts the RMSE results of these fine-tuning methods on Cu and Sn datasets. Adapter fine-tuning performs worse because the equivariance is destroyed under the fine-tuning process. Readout fine-tuning is not as accurate as full-parameter fine-tuning because only the last few layers are retuned, which lacks some flexibility compared to full-parameter fine-tuning.
>
> **Table 1 The comparasion of four different finetuning methods.**
>
> ||MACE (Full-parameter)|MACE (Adapter)|MACE (Readout)|MACE (ELoRA)|
> |:-:|:-:|:-:|:-:|:-:|
> |Cu,E|0.6|4.4|5.6|**0.4**|
> |Cu,F|5.4|22.9|28.2|**4.4**|
> |Sn,E|4.9|27.6|30.8|**4.6**|
> |Sn,F|31.7|67.5|74.8|**29.2**|
> |Number of trainable parameters|723866|169296|2192|176666|
>
> **Q3: Number of parameters after using ELoRA.**
>
> A3: In ELoRA, the path-dependent weights can be merged into the model weights after training, just like in the original LoRA. Thus, it will not increase the number of the model's parameters. The last row of Table 1 records the number of trainable parameters of the four different fine-tuning methods.
>
> **Q4：The applicability on other SO(3)-equivarant MPNNs.**
>
> A4: We add experiments with ELoRA on SevenNet [1] (another representative SO(3)-equivariant MPNN), as shown in Table 2. The conclusions drawn from Table 2 are consistent with those from Table 1. Our proposed ELoRA can be adapted to other SO(3)-equivariant MPNNs in a user-friendly manner.
>
> **Table 2 The finetuning methods on SevenNet.**
>
> ||SevenNet (Full-parameter)|SevenNet (Adapter)|SevenNet (Readout)|SevenNet (ELoRA)|
> |------|:-----------------------:|:----------------:|:----------------:|:--------------:|
> |Cu,E|0.9|2.5|10.2|**0.8**|
> |Cu,F|12.8|32.1|153.2|**12.2**|
> |Sn,E|3.4|6.8|16.4|**3.0**|
> |Sn,F|74.1|117.0|190.5|**73.4**|
>
> [1] Park, Yutack, et al. "Scalable parallel algorithm for graph neural network interatomic potentials in molecular dynamics simulations."
>
> **Q5: Results analysis on inorganic dataset.**
>
> A5: The prediction errors of energy and forces should be considered jointly, as they are typically optimized together during training. We cannot evaluate based solely on either energy or forces. For a comprehensive analysis, we can equally combine energy and force RMSEs as a joint metric. Under this metric, ELoRA achieves the best accuracy on 9 out of 10 datasets, as Table 3 shows.
>
> **Table 3 Summed RMSE of Energy and Force.**
>
> ||NequIP|Allegro|DPA2|MACE|MACE (Full-parameter)|MACE (ELoRA)|
> |:-:|:-:|:-:|:-:|:-:|:-:|:-:|
> |SSE-PBE|42.7|48.8|51.7|31.7|17.2|**14.5**|
> |H2O-PD|28.0|OOM|25.2|109.6|20.5|**16.9**|
> |Ag$\cup$Au-PBE|86.1|98.1|20.2|403.6|21.9|**17.6**|
> |Al$\cup$Mg$\cup$Cu|86.3|58.9|21.2|50.6|12.9|**11.0**|
> |Cu|22.9|10.2|10.1|52.4|6.0|**4.8**|
> |Sn|80.4|45.8|58.5|/|36.6|**33.8**|
> |Ti|165.0|92.5|118.1|102.5|85.3|**79.1**|
> |V|100.4|86.3|94.9|154.6|78.9|**72.9**|
> |W|181.2|105.6|113.7|196.8|93.3|**83.4**|
> |HfO2|60.3|65.4|55.2|**17.0**|30.5|21.3|
>
> **Q6: Clarify of preseved equivariance in ELoRA.**
>
> A6: ELoRA is a parameter-efficient fine-tuning method. Its parameters can be merged into the base model after tuned, the network structure remains the same. Thus, the equivariance is preserved.
>
> ELoRA projects the equivariant features into a lower-dimensional space through equivariant operations. These projected features remain SO(3)-equivariant, as proved in Proposition 4.4.
>
> **Q7: The explanation of the pretraining dataset's sparsity and the materials' diversity.**
>
> A7: The diversity refers to the combinatorial diversity of material structures. For example, even small organic molecules composed of carbon（C）, hydrogen（H）, oxygen (O), and nitrogen (N) atoms can theoretically form up to $10^{60}$ possible structures.
>
> The sparsity refers to the fact that labeled data with ab initio accuracy is in a small number. Computing structures' quantum properties (e.g., energy and atomic forces) require expensive Density Functional Theory (DFT) calculations. Currently, only a small fraction structures has high-quality DFT labels. The available labeled structures represent sparsity in the large structure space.
>
> **Q8: The organization of the paper.**
>
> A8: For a detailed explanation of the Section 3, please refer to Reviewer iY6i, Q1.

---

> > ### Comment · Reviewer_S1HM · 2025-04-03
> >
> > I appreciate the author's answer, and I would like to raise the score. But the author did not answer my question directly, hoping that the author will improve.
> >
> > >The readability of the paper is poor, especially in Section 3.
> > >* What is mean of L150-153: “They can only learn from the data (the dark blue circle in Figure 1(b)) in the specific downstream task they are trained on…”
> > >* Some terms should be standardized for academic expressions, e.g. complex downstream task data in L158 should refer to OOD data.
> > >* The organization of the paper is confusing, for example, the method mentioned in Section 3 (active learning/DP-GEN) does not seem to be relevant to this paper, and if it is, it should clarify the relationship with the existing works (limitation, improvement, etc.).

---

> > > ### Author Response · Authors · 2025-04-03
> > >
> > > Thank you again for your valuable comments. We apologize for not addressing the readability concerns more directly in our previous response due to the 5000 character limit.
> > >
> > > **Q1: The meaning of L150-L153.**
> > >
> > > A1: In this context, the model trained from scratch learns exclusively from the training set of the downstream task. In contrast, the pre-trained model benefits from training data across diverse tasks, enabling it to explore a more expansive configuration space. As shown in Figure 1, green dots represent the pre-training data, whereas orange dots correspond to the downstream task data. The model trained from scratch is trained exclusively on these orange dots, and thus can cover only the configuration space around them, indicated by the dark blue region in Figure 1(b). We will improve the description of Figure 1 in the revised version.
> > >
> > > **Q2: Academic expressions.**
> > >
> > > A2: We will revise the term "complex downstream task data" in Line 158 to "OOD data". We will also review the manuscript to ensure the consistent use of standardized academic expressions.
> > >
> > > **Q3: The organization of the paper.**
> > >
> > > A3: Section 3 serves as a transitional part and aims to convey the necessity of fine-tuning pre-trained models rather than training models from scratch under the consideration of generalization ability.
> > >
> > > We apologize for mentioning active learning in Section 3. Active learning, as mentioned in Section 3, could make people lose sight of this paper's main focus. In the revised version, we will remove the content related to active learning.
> > >
> > > In writing "active learning", we aimed to illustrate that active learning and pre-training fine-tuning are two main paradigms to enhance the generalization capability of models. Active learning improves model performance by iteratively labeling OOD data. Meanwhile, the pre-training–fine–tuning paradigm improves performance by leveraging large-scale pre-trained datasets.

---

### Official Review · Reviewer_qEUK · 2025-03-10

**Overall Recommendation:** 3

**Summary:**

Existing parameter-efficient fine-tuning (PEFT) methods are not suitable for equivariant GNNs. To that end, the authors propose a novel method for equivariant low-rank adaptation for finetuning equivariant GNNs. Specifically, the authors propose a path-dependent low-rank adaptation for the tensor product weights which preserves the equivariance of the operation. The authors evaluate their PEFT method compared to full-parameter fine-tuning and from-scratch training on several common organic and inorganic datasets.

**Claims And Evidence:**

The authors first provide a study of the singular-value decomposition of weight matrices from the fine-tuned model and the model trained from scratch and compare with the pretrained model. It is not clear to me how this supports the rest of the paper.

The main theoretical claims in Section 4 are supported with proofs in the appendix, which appear to be correct and are convincing.

However, I am not necessarily convinced by the empirical performance of ELoRA. Specifically, it is not clear that ELoRA consistently outperforms full-parameter finetuning based on the experiments provided. In fact, in Section 5.3.1, the authors actually show that given even a small amount of finetuning data (1000 samples), ELoRA and full-parameter fine-tuning perform the same. The inorganic dataset experiment provided provide support for the proposed method, but the organic dataset experiments do not.

The authors also make several claims about ELoRA reducing the number of parameters. In my view is a misleading claim, though I am less familiar with existing literature on parameter-efficient fine-tuning. After finetuning the model, the total number of parameters is the same.

**Essential References Not Discussed:**

The paper is currently missing the state-of-the-art method on all of the organic datasets provided [1].

[1] Equivariant Graph Network Approximations of High-Degree Polynomials for Force Field Prediction, Xu et al, https://arxiv.org/abs/2411.04219

**Experimental Designs Or Analyses:**

As mentioned previously, there are several issues with the organic dataset evaluations. The authors only train on a subset of the rMD17 dataset, which does not provide a fair comparison for their method. They also do not provide results for full-parameter finetuning on the 3BPA and AcAc datasets, making it impossible to evaluate ELoRA.

The first analysis on SVD of weight matrices is also confusing to me. Clearly, we should expect the weights of a fine-tuned model to have some similarity to it's original pre-trained weights, but even for separate training runs of the same model on the same dataset, is there any reason to expect that there would be similarity in the weight matrices? In my view, this analysis is not convincing and does not support the rest of the paper.

**Methods And Evaluation Criteria:**

The inorganic datasets used for evaluation are reasonable and provide a solid evaluation of ELoRA compared to full-parameter finetuning. Specifically, the authors demonstrate that ELoRA outperforms full-parameter finetuning even given ~10,000 finetuning samples, which provides strong support for the proposed method.

However, there are several issues with the evaluations on organic datasets. The only organic dataset used in the main paper is rMD17, and the authors choose to only train on 50 molecules, which is an unfair limitation for baseline methods. The full rMD17 dataset contains 100,000 structures for each molecule, and it is common for works to train on only 1,000 structures per molecule, however, as demonstrated in 5.3.1,  ELoRA does not outperform full-parameter fine-tuning with 1,000 training structures per molecule. They authors additionally evaluate on 3BPA and AcAc datasets in the appendix, however, they do not provide results for full-parameter finetuning, which makes it impossible to evaluate ELoRA on these tasks. Additionally, for rMD17, 3BPA, and AcAc, the state-of-the-art method, PACE [1] is not included in the comparison.

[1] Equivariant Graph Network Approximations of High-Degree Polynomials for Force Field Prediction, Xu et al, https://arxiv.org/abs/2411.04219

**Other Comments Or Suggestions:**

Unless the authors can significantly revise the study on SVD to better support the story of the paper, I think it would be better off if this section was moved to the appendix to make room for other experimental results (preferably on 3BPA and AcAc) in the main paper. It may also be interesting to compare the similarity of ELoRA vs. full-parameter finetuning with the pre-trained weights instead of fine-tuning vs. from scratch.

### Updates After Rebuttal

While I still feel that the rMD17 experiments are not so strong or realistic, at least the are following what was done by previous works. The rest of the experiments are strong, and the ideas in this paper are a good contribution to the MLFF community. I am leaning towards acceptance on this paper given the authors' response to all reviewers during the rebuttal period.

**Other Strengths And Weaknesses:**

Strengths:
- To my knowledge, ELoRA is the first work to enable parameter-efficient fine-tuning while preserving equivariance
- ELoRA demonstrates consistent performance improvement compared to full-parameter fine-tuning on several inorganic datasets

Weakness:
- The evaluation on organic datasets does not show that ELoRA outperforms full-parameter fine-tuning, making the experimental results of the paper as a whole inconclusive. Specifically, the study provided in Section 5.3.1 actually weakens the authors claims.
- The study on SVD of weight matrices does not support the rest of the paper.

**Questions For Authors:**

1) Can the authors provide results for ELoRA training on the standard split of 1,000 structures for each molecule on rMD17?
2) Can the authors provide results for full-parameter fine-tuning on 3BPA and AcAc?
3) Can the authors clarify the SVD experiments and how these results support the rest of the paper?

**Relation To Broader Scientific Literature:**

This work provides a novel method for parameter efficient finetuning of equivariant GNNs. Prior works have shown powerful results from finetuning pretrained equivariant foundation models, so this work is important for downstream applications.

**Theoretical Claims:**

I briefly checked the proofs of theoretical claims in the appendix and they appear to be correct, however it is possible that I may have missed some details.

---

> ### Author Rebuttal · Authors · 2025-04-01
>
> Many thanks to your questions and suggestions.
>
> **Q1: The fairness in using only 50 samples in rMD17 experiments.**
>
> A1: In practical downstream applications, fine-tuning methods are often expected to perform well with limited first-principles training data, as DFT calculations  are computationally expensive. To simulate such realistic requirements, we construct low-data scenarios (e.g., extract only 50 training samples from each small dataset in rMD17) to demonstrate the accuracy performance of dedicated models (ACE, NequIP) and fine-tuned pretrained model such as MACE.
>
> **Q2: Full-parameter finetuning results of 3BPA and AcAc.**
>
> A2: Our experiments show that ELoRA based MACE is better than full-parameter fine-tuned MACE, especially in the out of domain test set. Table1&2 show the from-scratch-training and finetuning reults of 3BPA and AcAc.
>
> **Table1 3BPA results**
>
> ||MACE (From scratch)|MACE (Full-parameter)|MACE (ELoRA)|
> |:-:|:-:|:-:|:-:|
> |300K,E|3.0 (0.2)|3.3 (0.03)|**3.0 (0.05)**|
> |300K,F|8.8 (0.3)|7.8 (0.01)|**7.5 (0.05)**|
> |600K,E|9.7 (0.5)|7.3 (0.04)|**6.5 (0.10)**|
> |600K,F|21.8 (0.6)|16.6 (0.05)|**15.5 (0.12)**|
> |1200K,E|29.8 (1.0)|20.3 (0.17)|**17.6 (0.11)**|
> |1200K,F|62.0 (0.7)|48.7 (0.56)|**42.0 (0.51)**|
> |dih,E|7.8 (0.6)|7.3 (0.28)|**5.9 (0.28)**|
> |dih,F|16.5 (1.7)|12.3 (0.10)|**11.4 (0.17)**|
>
> **Table2 AcAc results**
>
> ||MACE (From scratch)|MACE (Full-parameter)|MACE (ELoRA)|
> |:-:|:-:|:-:|:-:|
> |300K,E|0.9 (0.03)|1.0 (0.02)|**0.8 (0.03)**|
> |300K,F|5.1 (0.10)|5.1 (0.07)|**4.5 (0.06)**|
> |600K,E|4.6 (0.3)|5.8 (0.28)|**3.9 (0.33)**|
> |600K,F|22.4 (0.9)|16.4 (0.70)|**13.6 (0.26)**|
>
> **Q3: The comparison with SOTA results PACE.**
>
> A3: We will cite the PACE paper and include the results of PACE in the organic experiments. PACE represents SOTA results among dedicated models that are individually trained for each small dataset. However, our work focuses on developing novel fine-tuning techniques for pretrained models.
>
> In the case of rMD17, PACE achieves high accuracy when 1000 samples are used in training, as the PACE paper shows. In low-data scenarios, when training the PACE model with 50 samples per dataset, the accuracy cannot be reached, which is achieved with 1000 training samples. While, ELoRA makes it possible to obtain high-precision downstream models when a small amount of training data is provided.
>
> Table3 lists PACE and MACE results. The second and third columns show the MAE for PACE with 1000 and 50 training samples respectively. The fourth column reports the MAE results of MACE ELoRA. Table 3 reveals that PACE requires abundant training data to maintain its high accuracy. Its performance will drop at 50 training samples. ELoRA maintains prediction precision even with this minimal training set.
>
> **Table3 rMD17 results**
>
> ||PACE (1000, From scratch)|PACE (50, From scratch)|MACE (50, ELoRA)|
> |:-|:-:|:-:|:-:|
> |Aspirin,E|1.7|15.7|7.3|
> |Aspirin,F|5.8|37.4|17.6|
> |Azobenzene,E|0.5|6.7|4.0|
> |Azobenzene,F|2.2|17.5|12.4|
> |Benzene,E|0.02|0.6|0.2|
> |Benzene,F|0.2|3.3|1.6|
> |Ethanol,E|0.3|6.3|2.1|
> |Ethanol,F|1.8|25.4|10.7|
> |Malonaldehyde,E|0.5|11.5|6.5|
> |Malonaldehyde,F|3.6|57.3|21.7|
> |Naphthalene,E|0.2|2.1|1.4|
> |Naphthalene,F|0.9|9.7|6.0|
> |Paracetamol,E|0.9|10.1|4.8|
> |Paracetamol,F|4.0|29.3|14.8|
> |Salicylicacid,E|0.5|7.0|3.2|
> |Salicylicacid,F|2.9|29.2|14.2|
> |Toluene,E|0.2|2.7|1.3|
> |Toluene,F|1.1|12.0|5.9|
> |Uracil,E|0.3|5.9|2.1|
> |Uracil,F|2.0|26.8|11.6|
>
> **Q4: The analysis on SVD decomposition of weight matrices.**
>
> A4: Please refer to Reviewer iY6i, Q1.
>
> **Q5: Reduction in the number of parameters.**
>
> A5: The pretrained model covers multiple elements spanning the periodic table. But for downstream tasks, we retain only weights relevant to the task's elements and prune all others. This causes the reduction of model parameters. We will clarify this issue in the revised paper.

---

> > ### Comment · Reviewer_qEUK · 2025-04-01
> >
> > I thank the authors for their response to my comments. I have responded to their response below:
> >
> > >DFT calculations are computationally expensive. To simulate such realistic requirements, we construct low-data scenarios (e.g., extract only 50 training samples from each small dataset in rMD17)
> >
> > I do not feel that it is overly burdensome for practitioners to run at least 1000 DFT calculations when building a fine-tuning dataset, especially for small molecules as in rMD17. I maintain my claim that it is unreasonably restrictive to limit training to only 50 training samples.
> >
> > >Q2: Full-parameter finetuning results of 3BPA and AcAc.
> >
> > I thank the authors for providing these experiments and recommend that they include the results in their revised manuscript; this provides additional support for ELoRA.
> >
> > >Q4: The analysis on SVD decomposition of weight matrices.
> >
> > I still feel that the current SVD analysis does not provide any value. There is no reason to expect weights from different training runs to have any spectral similarity, and as such I think this is a misleading study. I recommend the authors move this section to the appendix and include the results from Q2 in the main paper.
> >
> > I appreciate the full-parameter finetuning results on 3BPA and AcAc. I think this provides stronger support for the proposed method. I still feel that the rMD17 results are not a realistic task and do not provide strong support for ELoRA, but I think that the overall idea of the paper and the other experiments are strong enough for me to raise my score.

---

> > > ### Author Response · Authors · 2025-04-02
> > >
> > > Thank you again for your thoughtful response.
> > >
> > > **Q1: 50 training samples for the rMD17 dataset.**
> > >
> > > A1: Fewer samples mean lower DFT computational costs. The rMD17 dataset has a relatively small number of atoms per sample, making the DFT computational cost not that expensive. However, when the number of atoms in a sample is large (e.g., over 300), the DFT computational cost increases substantially. We have some datasets with large atoms, but they may not be as representative as publicly available datasets, such as rMD17. Therefore, we choose the rMD17 dataset and use only a small number of samples to evaluate the fine-tuning performance of ELoRA.
> > >
> > > The required number of training samples depends on the system's complexity. We cannot use a specific quantity to define "large" or "small" number of samples, as this is highly system-dependent. To our knowledge, in the rMD17 dataset, 1000 training samples are considered a relatively large number and no more than 1000 samples should be used for training [1]. While for some complex systems, 1000 training samples may be insufficient.
> > >
> > > In Section 5.3.1, "Data Efficiency," we address the issue that when the number of training data in rMD17 increases to 1000 ("large"), the accuracy of full-parameter fine-tuning becomes comparable to ELoRA. Even with 50 ("small") training samples, ELoRA can achieve high accuracy. It could be inferred that in other complex systems, ELoRA may require only a "small" number of high-accuracy samples.
> > >
> > > In the MACE paper [2], the rMD17 dataset is trained with 50 samples, which demonstrates its data efficiency. In our rMD17 experiments, we followed their setting by adopting 50 training samples.
> > >
> > > [1] Revised MD17 dataset (rMD17)
> > > https://figshare.com/articles/dataset/Revised_MD17_dataset_rMD17_/12672038
> > >
> > > [2] Batatia, Ilyes, et al. “MACE: Higher order equivariant message passing neural networks for fast and accurate force fields.”
> > >
> > > **Q2: SVD Analysis.**
> > >
> > > A2: We will move the SVD analysis to the appendix and include AcAc and 3BPA results to the main paper to provide stronger supports for ELoRA.

---

### Official Review · Reviewer_iY6i · 2025-03-12

**Overall Recommendation:** 5

**Summary:**

The paper introduces a variant of LoRA for finetuning geometric graph neural networks that use spherical harmonics. The idea is to consider the main model parameters that appear in path-dependent tensor-product using Clebsch-Gordon in these SO(3) equivariant models, and provide path-dependent low-rank adaptation, which is shown to preserve equivariance. The proposed method is applied to fine tuning of pretrained MACE on organic and inorganic datasets for force-field prediction and it is compared against models trained from scratch or fully finetunned on the downstream dataset.

**Claims And Evidence:**

The claims on effectiveness of the proposed method is well-supported by experiments. Theoretical claims also make sense.

The paper’s original claim on providing low-rank adaptation for equivariant MPNNs at large is too broad. It should clarify at the outset that it is suggesting a method for fine-tuning that applies to methods using parameterized tensor-product using CG.

There is a claim at the beginning of the paper about the closeness of fine-tuned model to the pretrained model when compared against a model trained from scratch. The claim generally make sense, but the conclusion drawn from experiments is not accurate, since it is comparing two deep networks as functions based on the closeness of their weights’ spectra. It makes sense to acknowledge that difference in the weight space does not imply difference in the function space.

**Essential References Not Discussed:**

Nothing comes to mind.

**Experimental Designs Or Analyses:**

To the extent of my familiarity with this domain the experiments and the supporting analysis make sense. I appreciated the ablation on the rank.

**Methods And Evaluation Criteria:**

Please see my question on a design choice in the method.

I cannot comment on the specifics of experiments such as the choice of dataset for finetuning, as I am not familiar with those details, but the evaluation is in my view quite extensive and supportive.

**Other Comments Or Suggestions:**

Use of the term equivariant by the paper is confusing: vanilla message passing neural networks are also equivariant, but to symmetric group only. The paper is targeting the SO(3), and it makes sense to use better terminology to make the distinction. The term geometric GNN or MPNN, as opposed to equivariant MPNN, makes more sense.

**Other Strengths And Weaknesses:**

Strengths:

– The method is highly motivated given the extensive use of deep learning in molecules and materials and the need for finetuning on smaller datasets

– presentation is quite polished and the organization helps with a smooth delivery

– the paper shows perspective in discussing the problem and related literature

– experimental results are extensive and supportive

– the paper makes a good use of figures

Weakness:

– the scope of the contribution needs to be clarified early on, in the abstract and introduction.

**Questions For Authors:**

While the method generally makes sense, one design choice appears rather suboptimal: Given W is third order (ignoring channels) why this particular choice of decomposition for the tensor W? For example, why not use Tucker decomposition?

**Relation To Broader Scientific Literature:**

The paper is nicely positioned in the context of broader literature. In particular, literature on finetuning in geometric deep learning and important related works on neural networks for interatomic potentials are reviewed.

**Theoretical Claims:**

Theoretical claims of the paper are statements that appear to be correct, even without considering a formal proof. I have not checked the proofs.

---

> ### Author Rebuttal · Authors · 2025-04-01
>
> We thank the reviewer for the professional and valuable comments.
>
> **Q1: The analysis on SVD decomposition of weight matrices.**
>
> A1: We apologize for the analysis of the SVD decomposition, which could be misleading. Section 3 serves as a transitional part, aiming to convey the necessity of fine-tuning on pre-trained models rather than training models from scratch. Then, we naturally introduce our innovative fine-tuning method, ELoRA.
>
> In Section 3, Figure 1 provides a qualitative interpretation from the chemical space perspective, comparing the coverage function space among pre-trained models, fine-tuned models, and models trained from scratch. The SVD decomposition analysis intends to offer quantitative information, demonstrating that the weights of fine-tuned models exhibit higher similarity to the pre-trained models than those trained from scratch models.
>
> The function space represented by deep neural networks is complex due to its high dimension and nonlinear nature. To our knowledge, we have not yet found an alternative theoretical explanation to better characterize the learned function spaces. As a result, we use spectral data in each layer to figure out the knowledge embedded in model weights. It may not be a rigorous analysis. If the reviewers consider the SVD analysis insufficiently compelling, we can move the SVD experiments to the Appendix and move the AcAc and 3BPA experiments to the main text.
>
> As for spectral analysis on ELoRA-fine-tuned weights ($W_{\text{ELoRA}}$), we add the comparison on the pre-trained weights $W_0$ and $W_{\text{ELoRA}}$ (see Link: https://anonymous.4open.science/r/ELoRA/picture/spectra.png). The results show that $W_0$ and $W_{\text{ELoRA}}$ maintain high similarity. The distribution of cosine similarity is close to that of $W_0$ and full fine-tuned weights ($W_{\text{Full-parameter}}$). The added figure indicates that $W_{\text{ELoRA}}$ and $W_{\text{Full-parameter}}$ have high similarity to $W_0$.  We will add this figure in the revised manuscript.
>
> **Q2: The claim of "equivariant MPNNs"**
>
> A2: Our proposed fine-tuning method applies specifically to SO(3)-equivariant MPNNs. We will clarify the scope by specifying SO(3)-equivariant MPNNs in our revised paper.
>
> **Q3: Choice of decomposition.**
>
> A3: In Equation (8) $W^0_{l_3 l_2 l_1} + \Delta W^0_{l_3 l_2 l_1} = W^0_{l_3 l_2 l_1} + B_{l_3 l_2 l_1} A_{l_3 l_2 l_1}$, the weight matrix $W_{l_3 l_2 l_1}$ has dimensions $K^3_{l_3}$, $K^2_{l_2}$, and $K^1_{l_1}$. We merge $K^2_{l_2}$ and $K^1_{l_1}$ because the computation involves a transformation from an intermediate tensor of dimension $K^2_{l_2} \cdot K^1_{l_1}$ to an output tensor of dimension $K^3_{l_3}$, making this merging a natural design choice, as illustrated in Figure 3. There exist various weight decomposition method, identifying other more effective weight decomposition strategies (such as Tucker decomposition) will be one of our future works.

---

### Decision · Program_Chairs · 2025-05-01

**Decision:**

Accept (poster)

**Comment:**

The submission proposes an equivariant Low-Rank Adaptation (ELoRA) method for efficiently finetuning tensor-product-based equivariant GNNs. This method fills an important gap for finetuning equivariant models, and makes a reasonable contribution to the field of equivariant foundation models for molecular systems. The empirical demonstrations are quite supportive in general.

A few issues and queries are also raised during the review. Reviewer qEUK noticed the results on rMD17 are not under a fully convincing setting, which turns out not a critical problem after authors' explanation that it follows previous settings (the advantage under this setting is still not as significant as in the inorganic case, though). The authors properly responded to Reviewer S1HM's request for an empirical verification of equivariance and presentation issues, and Reviewer G2Qy's request of additional ablation study and hyperparameter choices. I didn't find remaining major problems. Considering the timely contribution, I would like to recommend an accept.